# BELLA: Black-box model Explanations by Local Linear Approximations

## Abstract

Understanding the decision-making process of black-box models has become not just a legal requirement, but also an additional way to assess their performance. However, the state of the art post-hoc explanation approaches for regression models rely on synthetic data generation, which introduces uncertainty and can hurt the reliability of the explanations. Furthermore, they tend to produce explanations that apply to only very few data points. In this paper, we present BELLA, a deterministic model-agnostic post-hoc approach for explaining the individual predictions of regression black-box models. BELLA provides explanations in the form of a linear model trained in the feature space. BELLA maximizes the size of the neighborhood to which the linear model applies so that the explanations are accurate, simple, general, and robust. BELLA can produce both factual and counterfactual explanations.

## 1 Introduction

Machine Learning (ML) and Artificial Intelligence (AI) models have been employed to handle tasks in various domains, including justice, healthcare, finance, self-driving cars, and many more. Consequently, legislative regulations have been proposed to protect interested parties and control the usage of these models. One example is the General Data Protection Regulation of the European Union (Goodman and Flaxman, 2017), which stipulates *the right to an explanation* in situations where an AI system has been employed in a decision-making process. The AI act (European Parliament and Council of the EU, 2024), too, has stipulated the transparency of AI models according to the level of risk they pose.

The main issue is that many ML models are *black-box models*, i.e. one cannot easily understand how they arrive at a decision. This has led to the emergence of explainable Artificial Intelligence (xAI), a research field that aims to make black-box models human-understandable. In this paper, we are concerned with understanding regression models, i.e., models that make a numerical prediction. We are interested in explaining a given prediction of such a model *post-hoc*, i.e., after it has been produced. This is usually done by building an interpretable surrogate-model (e.g., a decision tree) that mimics the black-box model and that can be used to understand the prediction.

Numerous approaches have been proposed to build such surrogate models, in particular SHAP (Lundberg and Lee, 2017), LIME (Ribeiro et al., 2016), and MAPLE (Plumb et al., 2018). We review them in Section 2. To evaluate the surrogate models, several criteria have been proposed: we want the surrogate model to be *accurate*, i.e., to reflect the predictions of the black-box model; we want it to be *simple*, i.e., to use few features; we want it to be *robust*, i.e., giving similar explanations to similar data points; we want it to be *general*, i.e., applicable to many data points; and we want them to be *counterfactual*, i.e., to tell us how we have to modify the data point at hand to get a different prediction. We survey these desiderata in Section 3. We find that existing approaches tend to be good on some of these criteria, but never excel on all of them. This is not surprising, as the desiderata stand in obvious conflict: A simple surrogate model, e.g., risks being not very accurate, because usually accurate predictions can be made only by the type of complex models that we wish to explain in the first place.

Our key idea (which we present in Section 4) is to train a local linear model on the neighborhood of the data point that we wish to explain. This allows us to develop an approach called BELLA (Black-box model Explanations by Local Linear Approximations). We can show through extensive experiments (in Section 5) on a dozen datasets that BELLA beats all existing approaches across nearly all desiderata.

## 2 Related Work

Explainable AI has received much attention in the scientific literature (Beaudouin et al., 2020; Guidotti et al., 2018; Adadi and Berrada, 2018; Murdoch et al., 2019; Burkart and Huber, 2021; Hassija et al., 2024). In this paper, we are interested in *post-hoc approaches*, i.e., those that add interpretability to a given black-box model. Some of these approaches have been developed specifically for a given type of learners (such as Gat et al. (2022) for neural models). However, we are interested in *model-agnostic* approaches, i.e., those that can interpret any black-box model. Some model-agnostic approaches compute feature importance (Chen et al., 2018; Bang et al., 2021). However, these approaches do not allow explaining unseen data points. Hence, we focus on approaches that build a *surrogate model*, i.e., a model that mimics the black-box model but that is interpretable by design (e.g., a decision tree). While *global* methods provide an interpretation of the black-box model behavior on the whole space, *local* models provide an interpretation for a single data point. In this paper, we are interested *model-agnostic post-hoc local* explanations for *regression models*, i.e., we aim to provide an explanation for a given real-valued decision by any type of model for a given data point. We are thus not interested in approaches that work for classification only (Vo et al., 2022; Mothilal et al., 2020; Bui et al., 2022; Vo et al., 2023).

One approach to deal with regression models is to adjust the methods for classification models (such as LORE (Guidotti et al., 2019)), e.g., by discretization or clustering. However, this loses information and may require domain knowledge. Therefore several approaches have been developed to natively support both classification and regression models: SHAP (Lundberg and Lee, 2017) introduces a game theory approach to compute the contribution of each feature. The explanation applies to a single data point and it is given as a linear combination of the feature contributions. In order to improve computation time, AcME (Dandolo et al., 2023) computes feature contributions based on the perturbations based on data quantiles. LIME (Ribeiro et al., 2016) generates synthetic data points by feature perturbations. This yields a weighted neighborhood that is used to train a linear model, whose coefficients are then used as an explanation. However, both LIME and SHAP compute feature contributions in a projected, binary, space, which does not correspond to the original feature space. MAPLE (Plumb et al., 2018) addresses this problem and uses Random Forests to assign weights to the training examples. In this way, it forms a weighted neighborhood on which the explanation applies. SHAP, LIME, and MAPLE are direct competitors to our method BELLA, and we will see in our experiments that BELLA outperforms all of them on the quality of the explanations.

DLIME (Zafar and Khan, 2019) is a deterministic variant of LIME that provides stable and consistent explanations. However, it requires extensive manual input, as the user has to provide the number of clusters for the hierarchical clustering step, the number of neighbors for the KNN step of the method, and the length of the explanation. As such, DLIME is not well suited for regression tasks and was thus applied only to classification.

Another group of approaches computes *counterfactual* explanations. One such method (White and Garcez, 2019) uses the idea of *b-counterfactuals*, i.e., the minimal change in the feature to gauge the prediction of the complex model. This method applies only to classification tasks. Another work (Dandl et al., 2020) uses Multi-Objective Optimization to compute counterfactual explanations, both for classification and regression. Another work (Redelmeier et al., 2021) uses Monte Carlo sampling for the same purpose. However, all of these approaches can compute only counterfactual explanations, not both factual and counterfactual explanations like BELLA.

## 3 Preliminaries

**Goal.** We are given a tabular dataset $T \subset F_1 \times ... \times F_n$, where each $F_i$ is a set of *feature values*. We are also given a function $Y : T \to \mathbb{R}$ that yields, for each $x \in T$, a *target value* $y \in \mathbb{R}$. These target values may,

e.g., have been produced by a black-box model, in which case the target value is a *prediction.* Consider now one data point $x \in T$ with its target value $y$. We aim to compute an *explanation* in the following sense (Das and Rad, 2020):

**Definition 1** *An explanation is additional meta-information, generated by an external algorithm or by the machine learning model itself, to describe the feature importance or relevance of an input instance towards a particular output classification.*

If the target value was produced by a black-box model, we cannot be sure post-hoc that the features we identify really contributed to the computation of the target value (the model may just as well have thrown a dice, independently of any feature values). However, if several data points with these or similar feature values produce a similar prediction, we can use abductive reasoning to infer that these features may have contributed to the prediction, and that, hence, the features constitute an explanation. This is in fact common in the literature (Ribeiro et al., 2016; Lundberg and Lee, 2017; Radulovic et al., 2021; Ignatiev et al., 2019).

**Quality measures.** Several properties of "good" explanations have been proposed. Some of them, such as plausibility and accordance with prior beliefs, require human evaluation. Among the criteria that do not, we commonly find (Miller, 2019; Guidotti et al., 2018; Burkart and Huber, 2021; Molnar, 2018):

1. **Fidelity**: we want the value that the surrogate model explains to be close to value that the black-box model predicts.

2. **Simplicity**: we want the explanation to contain few features.

3. **Robustness**: we want similar data points to have similar explanations.

In addition, users tend to favor explanations that apply to many data points (Radulovic et al., 2021). This appears counter-intuitive, as we aim to explain only a single data point, no matter the others. And yet, it is easy to see that an explanation such as "You have a high risk of diabetes because your body mass index is 27, your A1C level is 7%, and your blood sugar level is 210mg/dL" is little satisfactory, as it allows no generalization. More helpful is to know that, *generally*, people with a body mass index larger than 25, an AIC level above 6.5%, and a blood sugar level of 200 mg/dL have a high risk of diabetes (Mayo-Clinic, 2023). We would thus like to have:

4. **Generality**: we want the number of data points to which an explanation applies to be large.

**Desiderata.** In addition to the above quality measures, there are also several criteria in the literature that either apply or don't apply to a given method of explanation. One of them is (Miller, 2018; Wachter et al., 2017):

5. **Counterfactuality**: the ability to provide a set of modifications to the data point at hand that would entail a change in the decision of the black-box model.

Counterfactuality is obviously of interest to a user who wishes not just to understand the prediction, but also to actively influence it (e.g., after having been attributed a high risk of diabetes based on the results of a blood test).

Several methods for post-hoc explainability use randomization to probe the black-box model. However, this entails that the same data can lead to different explanations, which introduces uncertainty for the user (Zhang et al., 2019; Slack et al., 2020). We thus have an additional desideratum:

6. **Determinism**: the avoidance of randomization steps

Finally, some methods (Plumb et al., 2018) propose explanations that take the form of a linear equation, which allows computing the predicted value from the feature values. This is a very attractive property, as the user can toy with the explanation and apply it also to neighboring data points. We thus have a desideratum that we call

7. **Verifiability**: the possibility to compute the predicted value from the feature values

Given this plethora of quality measures and desiderata, it is not surprising that no existing method (including our own) can satisfy all of them perfectly. However, we can at least show that our method ticks all desiderata, and outperforms existing methods across nearly all quality measures.

# 4  BELLA

We are given a tabular dataset $T$, a data point $x \in T$, and a real-valued target value $y$, and we aim to compute an explanation for this target value. Note that, different from other methods such as LIME and SHAP, we need as input only the dataset $T$, and not in addition the model that generated the dataset. This means that, different from SHAP and LIME, our method can explain any labeled dataset, whether it is produced by a model or comes from another source (e.g., housing prices from real estate data).

To explain the target value $y$, our idea is to find a linear equation $y \approx w_1 \cdot x_1 + w_2 \cdot x_2 + \cdots + w_l \cdot x_l + w_0$, where $w_i$ are real-valued regression coefficients and $x_i$ are feature values of $x$ in $T$. Such an equation tells the user (1) what the important features are and (2) how they can be used to compute the predicted value. To find this equation, BELLA proceeds in three steps (Algorithm 1):

1. Compute the distance of $x$ to the other points in $T$.

2. Conduct a linear search to find the best neighborhood of $x$, according to a defined metric.

3. Train a sparse linear model on that neighborhood, and propose this model as an explanation.

---

**Algorithm 1** BELLA

    **Input**: Dataset $T$ with labels $Y$
           Labeled data point $x \in T$
  1: $d \leftarrow \text{ComputeDistances}(x, T)$
  2: $L, N \leftarrow \text{NeighborhoodSearch}(x, T, d)$
  3: **return** $L, N$

---

**Step 1: Computing the distances.** To compute the neighborhood of the input data point, we need a distance measure. A good starting point is to have all numerical features on the same scale so that each feature contributes to the distance measure in the same range. Therefore, we first standardize all numerical features to have a mean of 0 and a standard-deviation of 1.

To compute the distances, we employ the generalized distance function (Harikumar and Surya, 2015), which consists of three separate distance measures to account for numerical, categorical, and binary data types, as follows:

$$d(x, x') = \sum_{i=1}^{m_{num}} d_{num}(x_i, x_i') + \sum_{j=m_{num}+1}^{m_c+m_{num}} d_c(x_j, x_j') +$$
$$\sum_{k=m_c+m_{num}+1}^{m_{num}+m_c+m_b} d_b(x_k', x_k') \tag{1}$$

Here, $m_{num}$, $m_c$ and $m_b$ are the number of numerical, categorical, and binary features, respectively. The distance measure for the numerical attributes $d_{num}$ is the $L1$ norm $d_{num}(x, x') = |x - x'|$, which is preferred over $L2$, as it is more robust to outliers (Hopcroft and Kannan, 2014). For categorical features, $d_c$ is the distance measure (Ahmad and Dey, 2007), which takes into account the distribution of values and their co-occurrence with values of other attributes. The distance between two values $a$ and $a'$ of an attribute $A_i$ with respect to attribute $A_j$ is given by:

$$d_c^{ij}(a, a') = P(A_j \in \omega | A_i = a) + P(A_j \notin \omega | A_i = a') - 1$$

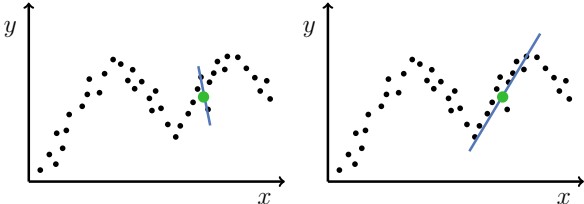

Figure 1: Left: an explanation for a data point $x$ that is too specific, applying only to a very small neighborhood. Right: An explanation that applies to a larger neighborhood, which is what we aim at.

Here, $P(A_j \in \omega | A_i = a)$ is the conditional probability that attribute $A_j$ will take a value from the set $\omega$ given that the attribute $A_i$ takes the value $a$. $\omega$ is a subset of all possible values of attribute $A_j$ that maximizes the sum of the probabilities. Since both probabilities can take values from $[0, 1]$, we subtract 1 in order to arrive at $d_c^{ij}(x, x') \in [0, 1]$. Lastly, for binary features, we use the Hamming distance: $d_h(x, x') = 1$ if $x = x'$, and zero otherwise. In Line 1 of Algorithm 1, the function *ComputeDistances* returns the distances by Equation 1. Note that our distance measure does not take the label into account. This is because there can be data points with different feature values and a similar prediction. In such cases, BELLA provides the explanation for the "correct" local neighborhood.

**Step 2: Neighborhood Search.** After computing the distances, we proceed with the exploration of the neighborhood of the input data point $x$. The goal is to find a set of points, closest to $x$ according to the distance measure, that will serve as a training set for a local surrogate model. Several common techniques could be considered to that end, including kNN, K-Means, and other distance-based clustering methods. In our case, however, we aim to find a neighborhood such that a linear regression model trained on that neighborhood represents an accurate local approximation of the black-box model. Hence, the quality of the neighborhood is proportional to the quality of the performance of the linear model fitted on it. Common drawbacks of regression evaluation metrics are missing interpretability, sensitivity to outliers and near-zero values, divisions by zero, missing bounds, and missing symmetry. We find that the *Berry-Mielke universal R value* $\Re$ (Berry and Mielke Jr, 1988) avoids most of these pitfalls. $\Re$ represents the measure of agreement between raters and it is a generalization of Cohen's kappa (Cohen, 1960). $\Re$ measures how much better the model is compared to a naive one (e.g., to a random predictor). $\Re$ takes values from the range $[0, 1]$, and it can be interpreted easily: If $\Re$ is equal to 0, the model performance is equal to the one of the random model and if it is 1, then the model has perfect performance. $\Re$ is defined as $\Re = 1 - \frac{\delta}{\mu}$, where $\delta$ and $\mu$ are defined as:

$$\delta = \frac{1}{n} \sum_{i=1}^{n} \Delta(\hat{y}_i, y_i), \quad \mu = \frac{1}{n^2} \sum_{i=1}^{n} \sum_{j=1}^{n} \Delta(\hat{y}_j, y_i). \tag{2}$$

Here, $n$ is the number of samples, $y_i$ is the actual target value, $\hat{y}_i$ is the predicted value, and $\Delta(\cdot)$ represents the distance function between the true and the predicted value. The original work by (Berry and Mielke Jr, 1988) uses the Euclidean distance, but later works (Janson and Olsson, 2001; 2004) propose to use the squared Euclidean distance instead, because this distance is equivalent to the variance of the variable, which further improves the interpretability of $\Re$. We follow this argumentation, and use $\Delta(a, b) = (a - b)^2$. This definition implies that $\Delta$ is in fact equal to the Mean Squared Error (MSE). Thus, by optimizing $\Re$, we are actually optimizing the accuracy of the local model.

However, to avoid explanations that are too specific, i.e., explanations that apply to very small neighborhoods, as in Figure 1 (left), we wish to optimize not just the *accuracy*, but also the *generality* of the surrogate model. Therefore, we include the size of the neighborhood in the optimization function, to aim for explanations that are at the same time accurate and general (Figure 1 (right)). One way to do this is to maximize the lower bound of the confidence interval of $\Re$. The lower and upper bounds for the confidence interval of $\Re$ are given by (Berry and Mielke Jr, 1988):

$$CI_\Re = \overline{\Re} \pm MOE_\Re = 1 - \frac{\overline{\delta} \mp MOE_\delta}{\mu} \tag{3}$$

Here, *MOE* stands for the Margin of Error. From Equation 3, it follows that computing the lower bound of $\Re$ is analogous to computing the upper bound of $\delta$. Therefore, we can compute the margin of error for $\delta$ as $MOE_\delta = t\frac{\sigma}{\sqrt{n}}$, where $\sigma$ is the standard deviation of the sample, $n$ is the sample size and $t$ represents the critical value from the t-distribution. We use the t-distribution because it is adapted for small sample sizes, which is what we encounter when we grow the neighborhood. The distribution converges to the normal distribution as the sample size increases.

Due to the non-monotonic nature of the $\Re$ value, we have to explore the whole space to maximize its lower bound. We employ a linear search algorithm (Algorithm 2) to this end.

---

**Algorithm 2** Neighborhood Search

    **Input**: Labeled data point $x \in T$
              Dataset $T$ with labels $Y$
              Distances $d\!:\!T \to \mathbb{R}$ of the data points to $x$
 1: Sort $T$ by ascending $d$
 2: $n \leftarrow$ number of features in $T$
 3: $max\Re_{lb} \leftarrow 0, bestN \leftarrow 0, bestL \leftarrow \emptyset$
 4: **for** $i = min(2n, |T|)$ to $|T|$ **do**
 5:     $L \leftarrow TrainLocalSurrogateModel(T[0:i])$
 6:     **if** $\Re_{lb}(L) > max\Re_{lb}$ **then**
 7:         $max\Re_{lb} \leftarrow \Re_{lb}(L), bestN \leftarrow i, bestL \leftarrow L$
 8:     **end if**
 9: **end for**
10: **return** *bestL, T[0:bestN]*

---

The algorithm receives as input a labeled data point $x$ that is to be explained, a labeled training set $T$, and a vector of distances between $x$ and each point in the training set $T$. We sort the training set by increasing distance to $x$, train a linear model on the first $i$ data points for increasing $i$, and return the set of neighbors for which the lower bound of $\Re$ is maximal. As the neighborhood is very small in the beginning, the training easily leads to overfitting. Therefore, we consider at least $2n$ data points for our neighborhood, where $n$ is the number of features. This ensures that the estimation of regression coefficients exhibits less than 10% relative bias (Austin and Steyerberg, 2015).

---

**Algorithm 3** Train Local Surrogate Model

    **Input**: Neighborhood of data points $N$
 1: $F \leftarrow$ the set of all features in $N$
 2: $F' \leftarrow \{f | f \in F \wedge VIF(f) < 10.0\}$
 3: $Features_{Lasso} \leftarrow Lasso(cv = 5, features = F')$
 4: **return** $OLS(Features_{Lasso})$

---

**Step 3: Building a local surrogate model.** We build our local surrogate model on the neighborhood we have found. To obtain a model with few parameters (i.e., a *simple* model), we use regularization. In terms of feature selection, $L1$ regularization (e.g. Lasso (Hastie et al., 2009)) is able to select a nearly perfect subset of variables in a wide range of situations. The only condition for this to work is that there are no highly collinear variables (Candès and Plan, 2009), which can significantly reduce the precision of estimated regression coefficients. To remove highly collinear features, we compute the *variance inflation factor* (VIF), and, following a rule of thumb (Stine, 1995), adopt 10 as the cut-off value for the VIF.

After removing highly collinear features, the next step is to train a linear model with Lasso regularization. Lasso regularization adds a penalty term in the form of the sum of absolute values of the regression coefficients. The objective function is $\min_{\beta \in \mathbb{R}^p}(||y - \beta X||_2^2 + \lambda||\beta||_1)$, where $\lambda$ is the shrinkage parameter. This provides a sparse model, by forcing some coefficients to be zero. Removing some features ensures a better generalization, and results in simpler, and thus more comprehensible explanations. On the other hand, coefficients obtained by minimizing the Lasso objective function are biased towards zero. Therefore, Lasso

Figure 2: Explanation example.

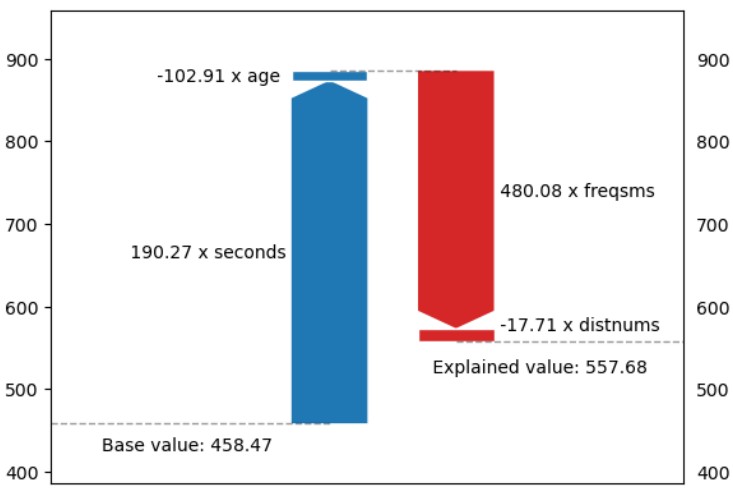

The value predicted by the model is **551** and the explained value is **557**. This explanation applies to **476** other instances.

is preferred for model selection rather than for prediction. The common strategy is to train an Ordinary Least Squares (OLS) linear model on the subset of variables selected by Lasso. This corresponds to a special variation of the relaxed Lasso (Meinshausen, 2007), with $\phi = 0$.

To determine the value of the shrinkage coefficient $\lambda$, we use 5-fold cross-validation (CV). To preserve the deterministic nature, we perform CV on adjacent slices of the dataset, without random shuffles. CV selects the best model in terms of the prediction error. Since the goal of this step is model selection, we want to avoid choosing $\lambda$ too small, and hence we apply the common one-standard error rule. According to this rule, the most parsimonious model is the one whose error is no more than one standard error above the error of the best model (Hastie et al., 2009).

Once we have obtained the most parsimonious model, i.e., the best set of features, we train the final local surrogate model as an OLS model using the features selected by Lasso. This procedure is described in Algorithm 3, and returns a local linear model.

This procedure is designed to maximize the robustness of our explanations: the Lasso regularization makes our results less brittle. Also, our target metric increases the number of data points, making the result more general and thus more robust.

**Providing an explanation.** As the final result, BELLA outputs the OLS model computed by Algorithm 3, together with the size of the neighborhood. As an example, consider the Iranian Churn dataset (Jafari-Marandi et al., 2020). It contains the (anonymized) customers of a telecommunication company, with their age, subscription length, satisfaction with the service, etc. The goal is to predict the commercial value of the customer to the company (in dollars).

Let us now consider a given customer, for which a black-box model predicted a commercial value of $551. The explanation that BELLA can provide for this prediction is shown in Figure 2. All numerical features have been standardized to have a mean value equal to 0 and a standard deviation equal to 1. (Thus, a customer has a "negative age" if they are younger than the average customer.) In the explanation, the base value is the output of the model when all inputs are set to zero (i.e. to their mean value). Each bar shows the total contribution of each feature to the predicted value – positive contributions are the blue upward arrows, and negative contributions are the red downward arrows. The more the customer phones (variable *seconds*), the more revenue the company generates. The age (which is below average for this particular customer), likewise, has a small positive impact. The number of SMS, in contrast, (variable *freqSMS*) impacts the revenue negatively. Finally, the number of distinct phone numbers called (variable *distnum*) has a small

negative impact. These sizes of the bars are easy to interpret: The size of each bar is equal to the value of the feature multiplied by the weight computed by our method. Their sum is then directly equivalent to the explained value:

$$y \approx 458.47 + 190.27 \times seconds - 102.91 \times age$$
$$-480.08 \times freqSMS - 17.71 \times distNums$$

This computation applies to all data points in the neighborhood of the input data point (to the current instance and 476 others in our example). We thus see that BELLA's explanations are *verifiable* (because they take the form of a linear equation), *deterministic* (because BELLA does not use any randomized steps), *simple* (because we applied regularization), *general* (because we maximized the neighborhood), and *accurate* (because we optimized the linear model on the local neighborhood). In addition, BELLA does not probe the black-box model. This means that, unlike many of its competitors, BELLA can explain not just the decisions of a black-box model, but any numerical variable in a tabular dataset – even if that variable was not generated by a model at all but merely observed in reality (such as, e.g., housing prices). Let us now turn to the missing desideratum, *counterfactuality*.

**Counterfactual explanations** provide information about a (minimal) change needed to alter the prediction of the black-box model. In a classification scenario, the goal is to make the model predict a different class. In a regression scenario, the goal is not to make the model predict *any other value*, but the value that the user would like to see. In our example of the Iranian churn dataset, an analyst may ask why the model predicted $551 instead of, say, $1000. A counterfactual explanation should suggest a set of changes that should be applied to reach this reference value. To provide such an explanation with BELLA, we select candidates, i.e. data points whose target value is in an $\epsilon$-vicinity to the reference value (with $\epsilon = 5\%$). There can be multiple candidates.

To find the best one, we optimize two criteria: the distance between the given data point and the candidate, and the amount of change needed. The first criterion will favor candidates that are in the vicinity of the given data point. The second criterion controls the amount of change applied. To alter the outcome, one can usually either apply a small change to several features, or a big change to few features. The second approach is risky: without human intervention, we can end up with a set of features that are difficult or impossible to change (e.g., the age of a customer). Therefore, we rather aim to minimize the average amount of change and suggest smaller adjustments to multiple features (such as frequency of use, or the number of SMS messages). This yields the following objective function:

$$\min_{x' \in T}(d(x, x') + \frac{d(x, \hat{x})}{len(Explanation)}) \tag{4}$$

Here, $x'$ is a counterfactual candidate data point, $d$ is a distance measure defined in Equation 1, $\hat{x}$ represents the modified data point $x$ according to the counterfactual explanation and *len(Explanation)* is the number of features that have been modified. The modified data point, $\hat{x}$, represents the counterfactual explanation.

The overall process is described in Algorithm 4. The algorithm takes as input the labeled dataset $T$, a labeled data point $x \in T$, a reference value $y_{ref} \in \mathbb{R}$, and a permitted deviation $\epsilon$ from the reference value. We first choose the set of counterfactual candidates $X'$ that have the target value in the $\epsilon$ neighborhood of the reference value $y_{ref}$. For each $x'$ among these candidates, we compute the explanation using BELLA[1]. This explanation gives us a set of features, and the proposed modification of $x$ is to set all these features of $x$ to the values given by $x'$. Among these proposed modifications, we choose the one that minimizes the objective function in Equation 4.

Such a counterfactual explanation may not always be possible: It can happen, for example, that there is no data point in the $\epsilon$ neighborhood. In such a case, the neighborhood has to be enlarged, leading to more remote example points. This is the price to pay for the fact that BELLA only ever produces real examples, and never synthetic examples. If the proposed counterfactual modifications do not lead to an actual outcome from the black-box within the desired target range, one can reduce the impact of Lasso regularization to keep even the smaller feature modifications.

---

[1]In principle, any algorithm that produces a linear model can be used in Algorithm 4.

---

**Algorithm 4** Computing a counterfactual explanation

---

**Input**: Dataset $T$ of data points $x_i$ with labels $y_i$
Labeled data point $x \in T$
A reference value $y_{ref} \in \mathbb{R}$
Deviation from reference value $\epsilon = 0.05$
1: $X' \leftarrow \{x_i \in T : y_i \in [y_{ref} - \epsilon, y_{ref} + \epsilon]\}$
2: **for** $x'_i \in X'$ **do**
3: $\quad L_i, N_i \leftarrow \text{BELLA}(T, x'_i)$ $\qquad\qquad\qquad\qquad\qquad\qquad\qquad\qquad\qquad$ ▷ Algorithm 1
4: $\quad \hat{x}_i \leftarrow x'_i$ *modified according to* $L_i$
5: **end for**
6: $x_{ref} \leftarrow argmin_{\hat{x}_i}(d(x, x'_i) + \frac{d(x, \hat{x}_i)}{len(Explanation)})$
7: **return** $x_{ref}$

---

## 5 Experiments

**Datasets.** We performed experiments on datasets from two standard repositories (Dua and Graff, 2017; Romano et al., 2021) (shown in Table 1). Among them is also a high-dimensional dataset, Superconductivity, with 81 features. All categorical features have been one-hot encoded and all numerical features have been standardized. We draw a random 10% of each dataset as testing data. To show that BELLA works with different families of models, we trained a random forest (with 100 trees), and a neural network (with one hidden layer with 500 nodes) as black-box models. Since the results do not differ much, we show only experiments with the neural network here, while the experiments with the random forest are in the supplementary material.

**BELLA.** Our method is implemented in Python. We set the step size to 10%. For the black-box models, we use the implementations of `scikit-learn` (Pedregosa et al., 2011). All experiments are run on a Fedora Linux (release 38) computer with an Intel(R) Xeon(R) v4 @ 2.20GHz CPU, a memory of 64 GB, and Python 3.9. All code and the data for BELLA and the experiments is available on Github (URL masked for anonymity).

**Competitors.** We compare BELLA to LIME (Ribeiro et al., 2016), SHAP (Lundberg and Lee, 2017) and MAPLE (Plumb et al., 2018). We use the implementations by the authors[2][3][4]. We do not compare to

---

[2]https://github.com/marcotcr/lime
[3]https://github.com/slundberg/shap
[4]https://github.com/GDPlumb/MAPLE/

Table 1: Regression Datasets

| Dataset | Features | Numerical | Categorical | Instances |
|---|---|---|---|---|
| Auto MPG | 7 | 6 | 1 | 392 |
| Bike | 12 | 9 | 3 | 8760 |
| Concrete | 8 | 8 | 0 | 1030 |
| Servo | 4 | 0 | 4 | 167 |
| Electrical | 12 | 12 | 0 | 10000 |
| Superconductivity | 81 | 81 | 0 | 21262 |
| White Wine Quality | 11 | 11 | 0 | 4898 |
| Real Estate Valuation | 5 | 5 | 0 | 414 |
| Wind | 14 | 14 | 0 | 6574 |
| CPU activity | 12 | 12 | 0 | 8192 |
| Echocardiogram | 9 | 6 | 3 | 17496 |
| Iranian Churn | 11 | 8 | 3 | 3150 |

Table 2: Fidelity comparison (RMSE – smaller is better)

| Dataset | Factual | | | | Counterf. |
| | LIME | MAPLE | BELLA | SHAP | BELLA |
| --- | --- | --- | --- | --- | --- |
| Auto MPG | $2.99_{\pm 0.830}$ | $\mathbf{0.86}_{\pm 0.270}$ | $1.45_{\pm 0.390}$ | $\mathbf{0.00}$ | $5.69_{\pm 1.210}$ |
| Bike | $579.76_{\pm 24.73}$ | $\mathbf{75.42}_{\pm 6.710}$ | $224.70_{\pm 12.90}$ | $\mathbf{0.00}$ | $1053.64_{\pm 4.25}$ |
| Concrete | $10.61_{\pm 1.410}$ | $\mathbf{2.13}_{\pm 0.290}$ | $4.87_{\pm 0.670}$ | $\mathbf{0.00}$ | $10.73_{\pm 1.86}$ |
| Servo | $0.75_{\pm 0.320}$ | $\mathbf{0.21}_{\pm 0.100}$ | $0.59_{\pm 0.260}$ | $\mathbf{0.00}$ | $1.12_{\pm 0.260}$ |
| Electrical | $0.02_{\pm 0.002}$ | $\mathbf{0.01}_{\pm 0.001}$ | $0.02_{\pm 0.002}$ | $\mathbf{0.00}$ | $\mathbf{0.02}_{\pm 0.003}$ |
| Supercond. | $23.17_{\pm 0.697}$ | $\mathbf{1.05}_{\pm 0.029}$ | $14.24_{\pm 0.434}$ | $\mathbf{0.00}$ | $57.50_{\pm 1.150}$ |
| White Wine | $0.36_{\pm 0.030}$ | $\mathbf{0.17}_{\pm 0.010}$ | $0.29_{\pm 0.020}$ | $\mathbf{0.00}$ | $1.53_{\pm 0.090}$ |
| Real Estate | $4.97_{\pm 1.170}$ | $\mathbf{1.48}_{\pm 0.540}$ | $2.01_{\pm 0.730}$ | $\mathbf{0.00}$ | $\mathbf{1.76}_{\pm 1.700}$ |
| Wind | $2.52_{\pm 0.382}$ | $\mathbf{1.15}_{\pm 0.173}$ | $1.69_{\pm 0.247}$ | $\mathbf{0.00}$ | $5.43_{\pm 0.753}$ |
| CPU Activity | $16.30_{\pm 1.620}$ | $\mathbf{0.81}_{\pm 0.060}$ | $1.18_{\pm 0.110}$ | $\mathbf{0.00}$ | $\mathbf{2.24}_{\pm 0.220}$ |
| Echocard. | $3.02_{\pm 0.046}$ | $\mathbf{1.82}_{\pm 0.031}$ | $2.84_{\pm 0.049}$ | $\mathbf{0.00}$ | $6.86_{\pm 0.101}$ |
| Iranian Churn | $172.13_{\pm 23.52}$ | $\mathbf{4.04}_{\pm 0.970}$ | $24.40_{\pm 7.950}$ | $\mathbf{0.00}$ | $\mathbf{30.49}_{\pm 3.770}$ |
| **Norm. avg.** | $0.10_{\pm 0.014}$ | $0.02_{\pm 0.004}$ | $0.05_{\pm 0.008}$ | $0.00$ | $0.14_{\pm 0.015}$ |

Table 3: Generality comparison (% - larger is better)

| Dataset | LIME | SHAP | MAPLE | BELLA |
| --- | --- | --- | --- | --- |
| Auto MPG | $21.12_{\pm 6.120}$ | $0.00$ | $\mathbf{60.09}_{\pm 9.030}$ | $44.21_{\pm 3.210}$ |
| Bike | $1.34_{\pm 0.140}$ | $0.00$ | $6.35_{\pm 0.075}$ | $\mathbf{45.08}_{\pm 2.020}$ |
| Concrete | $1.10_{\pm 0.420}$ | $0.00$ | $32.05_{\pm 1.100}$ | $\mathbf{34.02}_{\pm 4.140}$ |
| Servo | $13.43_{\pm 5.040}$ | $0.00$ | $\mathbf{76.23}_{\pm 6.330}$ | $75.28_{\pm 13.21}$ |
| Electrical | $0.02_{\pm 0.010}$ | $0.00$ | $8.44_{\pm 0.390}$ | $\mathbf{31.84}_{\pm 0.654}$ |
| Supercond. | $1.01_{\pm 0.071}$ | $0.00$ | $4.39_{\pm 0.070}$ | $\mathbf{52.78}_{\pm 0.124}$ |
| White Wine | $0.97_{\pm 0.024}$ | $0.00$ | $16.45_{\pm 0.340}$ | $\mathbf{33.68}_{\pm 3.075}$ |
| Real Estate | $3.43_{\pm 2.120}$ | $0.00$ | $\mathbf{47.89}_{\pm 3.790}$ | $39.37_{\pm 11.12}$ |
| Wind | $0.78_{\pm 0.190}$ | $0.00$ | $12.46_{\pm 0.540}$ | $\mathbf{100.0}_{\pm 0.000}$ |
| CPU Activity | $0.79_{\pm 0.130}$ | $0.00$ | $9.34_{\pm 0.215}$ | $\mathbf{30.36}_{\pm 1.780}$ |
| Echocard. | $0.06_{\pm 0.003}$ | $0.00$ | $9.19_{\pm 0.060}$ | $\mathbf{77.11}_{\pm 7.270}$ |
| Iranian Churn | $1.83_{\pm 0.210}$ | $0.00$ | $12.17_{\pm 0.410}$ | $\mathbf{30.21}_{\pm 2.890}$ |
| **Average** | $3.82_{\pm 1.206}$ | $0.00$ | $24.59_{\pm 1.863}$ | $\mathbf{49.50}_{\pm 4.119}$ |

methods that are designed for classification tasks, or that can provide only counterfactual explanations and not factual ones (see again Section 2).

## 5.1 Experimental results

We compare BELLA's performance against the competitors on the quality measures from Section 3. All tables show the average performance on the test set of each method with confidence intervals at $\alpha = 95\%$.

**Fidelity** is measured by the Root Mean Squared Error (RMSE) of the local surrogate models wrt. the predictions of the black-box models (Table 2, with a min-max normalized average). SHAP always has an error of 0. This is because it provides exact explanations that apply only to a single data point. Among the methods that apply to a neighborhood of points, MAPLE is constantly the best, followed closely by BELLA.[5] LIME comes last.

---

[5] The fidelity of BELLA could be improved by giving more weight to the explained examples, as MAPLE does, but this would compromise the advantage of BELLA of providing a linear explanation that is valid for the whole neighborhood with the same error margin.

Table 4: Simplicity comparison (smaller values are better). LIME requires the explanation size as input, and we give it the size of the explanation computed by BELLA.

| Dataset | SHAP | MAPLE | BELLA/LIME |
|---|---|---|---|
| Auto MPG | $9.00_{\pm 0.000}$ | $8.93_{\pm 0.090}$ | $\mathbf{3.90}_{\pm 0.310}$ |
| Bike | $11.54_{\pm 0.040}$ | $13.44_{\pm 0.080}$ | $\mathbf{8.47}_{\pm 0.110}$ |
| Concrete | $8.00_{\pm 0.000}$ | $8.00_{\pm 0.000}$ | $\mathbf{6.24}_{\pm 0.240}$ |
| Servo | $12.47_{\pm 0.320}$ | $17.88_{\pm 0.170}$ | $\mathbf{5.65}_{\pm 1.160}$ |
| Electrical | $12.00_{\pm 0.000}$ | $12.00_{\pm 0.000}$ | $\mathbf{9.40}_{\pm 0.184}$ |
| Supercond. | $70.29_{\pm 0.221}$ | $81.00_{\pm 0.000}$ | $\mathbf{14.19}_{\pm 0.182}$ |
| White Wine | $11.00_{\pm 0.000}$ | $11.00_{\pm 0.000}$ | $\mathbf{7.58}_{\pm 0.200}$ |
| Real Estate | $5.00_{\pm 0.000}$ | $5.00_{\pm 0.000}$ | $\mathbf{4.10}_{\pm 0.320}$ |
| Wind | $13.05_{\pm 0.213}$ | $14.00_{\pm 0.000}$ | $\mathbf{9.32}_{\pm 0.093}$ |
| CPU Activity | $12.00_{\pm 0.000}$ | $12.00_{\pm 0.000}$ | $\mathbf{9.56}_{\pm 0.190}$ |
| Echocardiogram | $7.33_{\pm 0.110}$ | $8.49_{\pm 0.121}$ | $\mathbf{7.07}_{\pm 0.090}$ |
| Iranian Churn | $9.14_{\pm 0.040}$ | $10.52_{\pm 0.060}$ | $\mathbf{4.76}_{\pm 0.160}$ |
| **Norm. Avg.** | $0.89_{\pm 0.004}$ | $0.96_{\pm 0.003}$ | $\mathbf{0.59}_{\pm 0.023}$ |

**Generality** is measured by the number of data points to which the explanation applies (as a percentage of all data points in the training set). One could give a hypercube for the size of the neighborhood, but it is arguably the number of data points (and not the size of a potentially sparse hypercube) that conveys the significance of the explanation. Thus, for BELLA, we simply return the size of the neighborhood. For MAPLE we return the number of data points that have weights larger than 0. For LIME, an explanation comes with the range of values for each feature. We count the number of data points that fall into this range. The results are shown in Table 3. For SHAP, the size of the neighborhood is always 0. This is because SHAP provides feature contributions that are specific for the given data point, and there is no way to apply these explanations to other data points. LIME's explanations are more general, and MAPLE's explanations even more. Still, they are vastly less general than the explanations of BELLA.

**Simplicity** is most commonly measured by the number of features that an explanation contains (Table 4). LIME has the same size of explanations as BELLA. This is because LIME takes this parameter as input and we set it to the size of the explanation provided by BELLA. SHAP and MAPLE constantly provide longer explanations than BELLA. MAPLE has higher complexity than SHAP, even though it comes with lower accuracy.

One could consider tuning the simplicity of LIME until LIME beats BELLA on fidelity, or tune fidelity until LIME beats BELLA on simplicity. To compare the two methods, however, one has to fix one parameter and compare the other. This is what our experiments do: at the same simplicity, BELLA beats LIME on fidelity (Table 2). It follows that, to achieve the same fidelity as BELLA, LIME necessarily has to decrease its simplicity. Thus, BELLA beats LIME in both cases.

**Robustness** judges how similar the explanations for close data points are. We measure robustness as:

$$robustness = 1 - \frac{1}{n} \sum_{i=1}^{n} \frac{|\beta_{1i} - \beta_{2i}|}{|\beta_{1i}| + |\beta_{2i}|}. \tag{5}$$

Here, $n$ is the number of features, and $\beta_{1i}$ and $\beta_{2i}$ are the weights of feature $i$ in the first and second explanation, respectively. Robustness is in the range of $[0, 1]$, with 1 indicating that two explanations are identical. We compute explanations for each data point in the test set, and compute robustness wrt. a smaller set of 5 closest neighbors, and a larger set of 20 closest neighbors (Table 5). As expected, all methods are a bit less robust when the set of neighbors is larger, but otherwise the results are very similar: LIME samples 5000 data points to create a synthetic neighborhood. Thus, LIME can perform better than our approach on datasets that have fewer observations. However, in the vast majority of cases, as well as on average, BELLA outperforms LIME. BELLA also outperforms SHAP by a wide margin. This is because SHAP's

Table 5: Robustness comparison (0 to 1 – larger is better)

| Dataset | Number of Neighbors = 5 | | | | Number of Neighbors = 20 | | | |
|---|---|---|---|---|---|---|---|---|
| | LIME | SHAP | MAPLE | BELLA | LIME | SHAP | MAPLE | BELLA |
| Auto MPG | $0.89_{\pm0.050}$ | $0.74_{\pm0.070}$ | $0.73_{\pm0.060}$ | $\mathbf{0.91}_{\pm0.040}$ | $0.82_{\pm0.053}$ | $0.69_{\pm0.065}$ | $0.66_{\pm0.034}$ | $\mathbf{0.85}_{\pm0.077}$ |
| Bike | $0.78_{\pm0.030}$ | $0.67_{\pm0.040}$ | $0.52_{\pm0.040}$ | $\mathbf{0.83}_{\pm0.050}$ | $\mathbf{0.82}_{\pm0.027}$ | $0.63_{\pm0.032}$ | $0.52_{\pm0.047}$ | $0.81_{\pm0.035}$ |
| Concrete | $0.79_{\pm0.050}$ | $\mathbf{0.81}_{\pm0.030}$ | $0.68_{\pm0.060}$ | $0.74_{\pm0.070}$ | $0.63_{\pm0.050}$ | $\mathbf{0.70}_{\pm0.047}$ | $0.64_{\pm0.044}$ | $0.64_{\pm0.058}$ |
| Servo | $\mathbf{0.85}_{\pm0.040}$ | $0.46_{\pm0.020}$ | $0.53_{\pm0.110}$ | $0.76_{\pm0.070}$ | $\mathbf{0.88}_{\pm0.339}$ | $0.42_{\pm0.313}$ | $0.55_{\pm0.161}$ | $0.78_{\pm0.041}$ |
| Electrical | $0.65_{\pm0.029}$ | $0.58_{\pm0.041}$ | $0.69_{\pm0.045}$ | $\mathbf{0.83}_{\pm0.031}$ | $0.62_{\pm0.015}$ | $0.55_{\pm0.022}$ | $0.74_{\pm0.034}$ | $\mathbf{0.83}_{\pm0.022}$ |
| Supercond. | $\mathbf{0.91}_{\pm0.017}$ | $0.80_{\pm0.036}$ | $0.60_{\pm0.076}$ | $\mathbf{0.91}_{\pm0.036}$ | $0.90_{\pm0.030}$ | $0.74_{\pm0.044}$ | $0.49_{\pm0.061}$ | $\mathbf{0.92}_{\pm0.044}$ |
| White Wine | $0.72_{\pm0.060}$ | $0.64_{\pm0.060}$ | $0.71_{\pm0.060}$ | $\mathbf{0.77}_{\pm0.070}$ | $0.59_{\pm0.041}$ | $0.54_{\pm0.042}$ | $0.64_{\pm0.040}$ | $\mathbf{0.67}_{\pm0.047}$ |
| Real Estate | $0.77_{\pm0.060}$ | $0.75_{\pm0.060}$ | $\mathbf{0.78}_{\pm0.060}$ | $\mathbf{0.78}_{\pm0.090}$ | $0.67_{\pm0.046}$ | $0.65_{\pm0.051}$ | $0.62_{\pm0.059}$ | $\mathbf{0.77}_{\pm0.044}$ |
| Wind | $0.68_{\pm0.059}$ | $0.59_{\pm0.039}$ | $0.66_{\pm0.028}$ | $\mathbf{0.99}_{\pm0.007}$ | $0.72_{\pm0.085}$ | $0.56_{\pm0.047}$ | $0.62_{\pm0.025}$ | $\mathbf{0.98}_{\pm0.016}$ |
| CPU Activity | $0.53_{\pm0.080}$ | $0.73_{\pm0.060}$ | $0.70_{\pm0.040}$ | $\mathbf{0.82}_{\pm0.050}$ | $0.42_{\pm0.038}$ | $0.71_{\pm0.050}$ | $0.75_{\pm0.025}$ | $\mathbf{0.81}_{\pm0.035}$ |
| Echocardiogram | $0.77_{\pm0.031}$ | $0.64_{\pm0.038}$ | $0.61_{\pm0.041}$ | $\mathbf{0.92}_{\pm0.057}$ | $0.76_{\pm0.037}$ | $0.63_{\pm0.036}$ | $0.57_{\pm0.033}$ | $\mathbf{0.84}_{\pm0.070}$ |
| Iranian Churn | $0.71_{\pm0.090}$ | $0.86_{\pm0.030}$ | $0.67_{\pm0.050}$ | $\mathbf{0.89}_{\pm0.060}$ | $0.64_{\pm0.059}$ | $0.80_{\pm0.040}$ | $0.60_{\pm0.055}$ | $\mathbf{0.86}_{\pm0.050}$ |
| **Average** | $0.75_{\pm0.050}$ | $0.69_{\pm0.044}$ | $0.66_{\pm0.056}$ | $\mathbf{0.85}_{\pm0.053}$ | $0.71_{\pm0.068}$ | $0.64_{\pm0.066}$ | $0.62_{\pm0.051}$ | $\mathbf{0.81}_{\pm0.045}$ |

Table 6: Execution time comparison (in seconds – lower is better)

| Dataset | LIME | SHAP | MAPLE | BELLA |
|---|---|---|---|---|
| Auto MPG | $2.21_{\pm0.043}$ | $0.04_{\pm0.001}$ | $\mathbf{0.02}_{\pm0.001}$ | $1.40_{\pm0.034}$ |
| Bike | $1.84_{\pm0.001}$ | $0.10_{\pm0.000}$ | $\mathbf{0.01}_{\pm0.033}$ | $2.70_{\pm0.000}$ |
| Concrete | $2.10_{\pm0.034}$ | $\mathbf{0.02}_{\pm0.001}$ | $\mathbf{0.02}_{\pm0.001}$ | $0.92_{\pm0.023}$ |
| Servo | $1.34_{\pm0.043}$ | $0.17_{\pm0.001}$ | $\mathbf{0.02}_{\pm0.001}$ | $2.99_{\pm0.071}$ |
| Electrical | $3.27_{\pm0.008}$ | $0.13_{\pm0.000}$ | $\mathbf{0.01}_{\pm0.000}$ | $2.27_{\pm0.001}$ |
| Superconductors | $21.9_{\pm0.014}$ | $0.17_{\pm0.020}$ | $\mathbf{0.02}_{\pm0.001}$ | $242.00_{\pm0.752}$ |
| White Wine | $3.71_{\pm0.053}$ | $0.16_{\pm0.002}$ | $\mathbf{0.02}_{\pm0.001}$ | $1.66_{\pm0.014}$ |
| Real Estate | $1.74_{\pm0.069}$ | $\mathbf{0.01}_{\pm0.000}$ | $\mathbf{0.01}_{\pm0.000}$ | $0.72_{\pm0.032}$ |
| Wind | $3.77_{\pm0.004}$ | $0.13_{\pm0.002}$ | $\mathbf{0.01}_{\pm0.000}$ | $1.53_{\pm0.001}$ |
| CPU Activity | $1.04_{\pm0.001}$ | $0.11_{\pm0.000}$ | $\mathbf{0.01}_{\pm0.000}$ | $1.36_{\pm0.001}$ |
| Echocardiogram | $1.83_{\pm0.003}$ | $\mathbf{0.01}_{\pm0.001}$ | $\mathbf{0.01}_{\pm0.000}$ | $3.35_{\pm0.002}$ |
| Iranian Churn | $2.83_{\pm0.037}$ | $0.04_{\pm0.001}$ | $\mathbf{0.02}_{\pm0.001}$ | $1.36_{\pm0.024}$ |
| **Median** | $2.16_{\pm0.024}$ | $0.11_{\pm0.001}$ | $0.02_{\pm0.001}$ | $1.6_{\pm0.019}$ |
| **Average** | $3.97_{\pm0.024}$ | $0.09_{\pm0.001}$ | $0.02_{\pm0.001}$ | $21.86_{\pm0.019}$ |

explanations are tailored for a single data point. BELLA also outperforms MAPLE. This is because the crisp neighborhood of BELLA provides much more robust explanations than MAPLE's weighted neighborhood.

From Tables 2, 3, 4, and 5, we can see that at the same level of simplicity, BELLA provides more general, more robust, and more accurate explanations than LIME. BELLA provides less accurate explanations than SHAP and MAPLE, but at the same time, BELLA's explanations are more general, more robust, and vastly simpler. The results when the black-box model is a random forest are shown in the appendix, and they do not differ much.

**Runtime** of all methods is shown in Table 6. On average, MAPLE and SHAP are extraordinarily fast, and LIME is slower. BELLA is, on average, 5× slower than LIME. This is due mainly to a single dataset, Superconductivity, which has a very large number of features, and on which BELLA is 11× slower than LIME. This is because, different from our competitors, BELLA is deterministic. Our method thus has to explore the whole local space. At the same time, BELLA runs in the same order of magnitude of time as LIME on average, and it remains thus competitive. On the median, BELLA is even faster than LIME.

**Counterfactuality** is a desideratum that only BELLA, and neither LIME nor SHAP nor MAPLE fulfills. To evaluate the quality of BELLA's counterfactual explanations, we measure the fidelity of an explanation wrt. a reference value. For each data point $x$ with its target value $y$, we set the reference values to

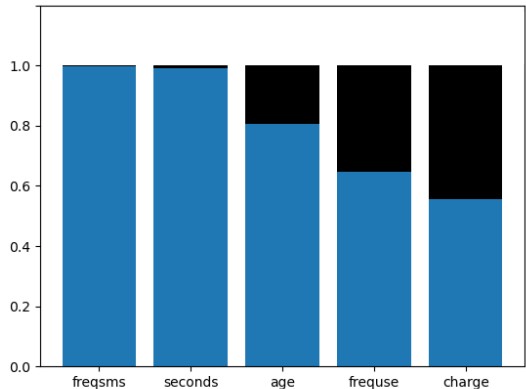

Figure 3: Top 5 features of a linear model on the Iranian Churn dataset, and percentage of data points whose BELLA explanation uses each feature.

$y_{ref} = y \pm 0.3 \times |y_{max} - y_{min}|$, so as to be different, but not unrealistically far away from $y$. The last column of Table 2 shows the RMSE of the counterfactual explanations (as well as of the factual explanations by BELLA and LIME for reference). We see that the counterfactual explanations of BELLA are often of similar fidelity as its factual explanations. Even in the cases where the error of counterfactual explanations is an order of magnitude larger, it is still lower or comparable to the error of the factual-only explanations provided by LIME.

**Other desiderata** outlined in Section 3 were determinism, and verifiability (the possibility to compute the explained value from the feature values). SHAP offers none of these. Neither does LIME. While both SHAP and LIME compute linear models with feature weights, these models are not verifiable in our sense: There is no way that the user can insert the feature values of a neighboring point into these models and obtain an explained value. This is because the linear models do not operate in the original input feature space. Only MAPLE offers this verifiability. However, it relies on randomization and provides no counterfactuality. BELLA is thus the only approach that delivers deterministic, verifiable, and both factual and counterfactual explanations.

**Verification on an interpretable model.** To confirm that the explanations provided by BELLA represent what the black-box model has learned, we evaluate them with regard to an already interpretable model. Instead of a black-box model, we train an Ordinary Least Square linear regression model and consider the 5 most important features. We then compute the explanations for each data point in the test set with our method. BELLA was able to recover on average 85.12% of the original top-5 features across all datasets. Figure 3 shows an example of the 5 most important features in the Iranian Churn dataset, as extracted by the linear model, and the percentage of data points for which BELLA gives an explanation with these features. This shows that our method provides explanations that generally agree with prior beliefs, as encoded in an interpretable model.

## 6 Limitations

BELLA is a domain-independent method to produce deterministic post-hoc local explanations of tabular regression datasets. This scope entails some inherent limitations: First, as a generalist method, BELLA does not take into account domain-specific peculiarities, such as domain-specific weighting, dynamic scaling of feature distances, or domain-specific distance functions. BELLA also currently does not take into account domain-specific lists of features that can or cannot be modified, so that it may produce de facto impossible counterfactuals (as in: "To avoid a cancer diagnostic, modify your age to be 10 years younger"). It may thus prove necessary to adapt BELLA to specific domains. While such adaptations are certainly possible, they are

out of the scope of the present paper. Second, BELLA aims at simple and verifiable (and ultimately linear) explanations. With this, BELLA may fail to capture intricate, higher-order interactions of features. This problem is part of the intrinsic trade-off between simplicity and accuracy, which any post-hoc explainability approach finds itself in. Third, BELLA has to explore the whole neighborhood space without sampling, because it is deterministic. This incurs a high algorithmic cost. In future work, this cost could be alleviated by using more efficient data structures (such as KD-trees). One could even give up on the desideratum of determinism, and resort to sampling. Such advances are, likewise, left for future work.

There are also a number of ethical considerations: First, BELLA does not correct for biases in the data or the model. If the data contains biased samples, discriminating features, or an otherwise unprofessional selection of features or data points, these characteristics will be mirrored in BELLA's explanations. We believe, however, that this is a feature rather than a bug: BELLA's explanations make such biases visible to the user. It would be disastrous to correct them and thus convey the impression that the model does not have them. A second concern is the problem of overtrust, where users mistakenly take BELLA's local explanation for a global correlation. That is an issue inherent to all local approaches, which would have to be countered by educating the user before using the approach. Finally, there is always the possibility that users "game" the model without genuinely improving underlying attributes (e.g., artificially adjusting specific features to increase a credit score). BELLA's counterfactual explanations might then recommend such tweaks also to other users.

For all of these reasons, BELLA, as any explainability method, gains from being combined with frameworks for fairness, transparency, and accountability, as well as domain-specific adaptations.

## 7 Conclusion

We have presented BELLA, an approach to provide post-hoc local explanations for any regression black-box model, or indeed any static tabular dataset with a numeric variable to be explained. BELLA is deterministic, and can provide both factual and counterfactual explanations. BELLA's objective function ensures accurate, general, robust, and simple explanations. Detailed experiments show that BELLA outperforms state-of-the-art approaches on these desiderata, often by a wide margin.

Future work could investigate methods to improve the speed of BELLA. One possibility is to use other data structures to speed up the neighborhood search. Another possibility is to give up determinism and resort to sampling to further speed up the algorithm. Finally, one could investigate even if BELLA could replace black box models for making a prediction in the first place, following positive experiences with linear models elsewhere (Ismail et al., 2022).

We hope that our work can open the door to research along this line and others, and ultimately make machine learning models more interpretable.

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

## A    Experiments with Random Forest as black-box model

In the main paper, we presented experiments using a neural network as a black-box model. Here, we show the results of experiments using a random forest of 100 trees as a black-box model. The results are shown in Tables 7, 8, 9, and 11. They do not differ much from the results on the neural network black-box model.

Table 7: Fidelity comparison for Random Forest as black-box model

| Dataset | Factual | | | | Counterf. |
| | LIME | MAPLE | BELLA | SHAP | BELLA |
| --- | --- | --- | --- | --- | --- |
| Auto MPG | $1.57_{\pm 0.470}$ | $\mathbf{0.95}_{\pm 0.272}$ | $1.52_{\pm 0.412}$ | $\mathbf{0.00}$ | $2.97_{\pm 0.880}$ |
| Bike | $338.50_{\pm 20.92}$ | $\mathbf{61.23}_{\pm 5.560}$ | $228.14_{\pm 14.48}$ | $\mathbf{0.00}$ | $1057.2_{\pm 3.782}$ |
| Concrete | $5.78_{\pm 0.844}$ | $\mathbf{2.22}_{\pm 0.341}$ | $5.11_{\pm 0.681}$ | $\mathbf{0.00}$ | $4.25_{\pm 0.942}$ |
| Servo | $0.45_{\pm 0.251}$ | $\mathbf{0.18}_{\pm 0.152}$ | $0.45_{\pm 0.211}$ | $\mathbf{0.00}$ | $4.47_{\pm 1.17}$ |
| Electrical | $0.01_{\pm 0.002}$ | $\mathbf{0.00}_{\pm 0.001}$ | $0.01_{\pm 0.002}$ | $\mathbf{0.00}$ | $\mathbf{0.00}_{\pm 0.004}$ |
| Supercond. | $30.68_{\pm 1.407}$ | $\mathbf{0.96}_{\pm 0.086}$ | $15.52_{\pm 0.481}$ | $\mathbf{0.00}$ | $57.50_{\pm 1.495}$ |
| White Wine | $0.30_{\pm 0.022}$ | $\mathbf{0.17}_{\pm 0.014}$ | $0.28_{\pm 0.023}$ | $\mathbf{0.00}$ | $0.87_{\pm 0.063}$ |
| Real Estate | $5.19_{\pm 1.292}$ | $\mathbf{2.55}_{\pm 0.962}$ | $5.04_{\pm 1.503}$ | $\mathbf{0.00}$ | $10.71_{\pm 2.632}$ |
| Wind | $1.40_{\pm 0.197}$ | $\mathbf{0.70}_{\pm 0.128}$ | $1.17_{\pm 0.184}$ | $\mathbf{0.00}$ | $3.84_{\pm 0.492}$ |
| CPU Activity | $12.57_{\pm 1.110}$ | $\mathbf{0.63}_{\pm 0.070}$ | $1.26_{\pm 0.010}$ | $\mathbf{0.00}$ | $2.54_{\pm 0.011}$ |
| Echocard. | $3.31_{\pm 0.510}$ | $\mathbf{1.64}_{\pm 0.246}$ | $3.21_{\pm 0.532}$ | $\mathbf{0.00}$ | $16.87_{\pm 1.869}$ |
| Iranian Churn | $141.57_{\pm 20.44}$ | $\mathbf{9.46}_{\pm 1.783}$ | $17.39_{\pm 2.923}$ | $\mathbf{0.00}$ | $320.59_{\pm 23.83}$ |
| **Norm. avg.** | $0.07_{\pm 0.010}$ | $0.02_{\pm 0.004}$ | $0.05_{\pm 0.008}$ | $0.00$ | $0.18_{\pm 0.026}$ |

Table 8: Generality comparison (% - larger is better)

| Dataset | LIME | SHAP | MAPLE | BELLA |
|---------|------|------|-------|-------|
| Auto MPG | $8.35_{\pm 2.342}$ | 0.00 | $45.41_{\pm 3.423}$ | $\mathbf{72.08}_{\pm 4.032}$ |
| Bike | $1.34_{\pm 0.247}$ | 0.00 | $6.36_{\pm 0.259}$ | $\mathbf{50.96}_{\pm 2.346}$ |
| Concrete | $0.23_{\pm 0.013}$ | 0.00 | $30.13_{\pm 1.894}$ | $\mathbf{42.24}_{\pm 4.038}$ |
| Servo | $11.03_{\pm 0.701}$ | 0.00 | $73.24_{\pm 6.034}$ | $\mathbf{84.03}_{\pm 12.123}$ |
| Electrical | $0.01_{\pm 0.001}$ | 0.00 | $7.23_{\pm 0.370}$ | $\mathbf{33.24}_{\pm 6.540}$ |
| Supercond. | $0.01_{\pm 0.002}$ | 0.00 | $3.47_{\pm 0.720}$ | $\mathbf{67.45}_{\pm 13.67}$ |
| White Wine | $2.18_{\pm 0.242}$ | 0.00 | $18.48_{\pm 0.373}$ | $\mathbf{72.24}_{\pm 2.439}$ |
| Real Estate | $2.43_{\pm 0.976}$ | 0.00 | $47.32_{\pm 3.987}$ | $\mathbf{83.38}_{\pm 7.320}$ |
| Wind | $0.44_{\pm 0.013}$ | 0.00 | $12.46_{\pm 0.440}$ | $\mathbf{100.0}_{\pm 0.000}$ |
| CPU Activity | $0.47_{\pm 0.014}$ | 0.00 | $9.94_{\pm 0.245}$ | $\mathbf{50.43}_{\pm 1.430}$ |
| Echocard. | $2.11_{\pm 0.440}$ | 0.00 | $5.98_{\pm 0.254}$ | $\mathbf{88.22}_{\pm 5.620}$ |
| Iranian Churn | $1.78_{\pm 0.336}$ | 0.00 | $11.97_{\pm 0.325}$ | $\mathbf{28.43}_{\pm 1.893}$ |
| **Average** | $2.53_{\pm 0.444}$ | 0.00 | $22.67_{\pm 1.3610}$ | $\mathbf{64.39}_{\pm 3.605}$ |

Table 9: Simplicity comparison (smaller values are better)

| Dataset | SHAP | MAPLE | BELLA/LIME |
|---------|------|-------|------------|
| Auto MPG | $9.00_{\pm 0.000}$ | $8.85_{\pm 0.210}$ | $\mathbf{3.65}_{\pm 0.170}$ |
| Bike | $12.22_{\pm 0.039}$ | $13.43_{\pm 0.081}$ | $\mathbf{7.94}_{\pm 0.103}$ |
| Concrete | $8.00_{\pm 0.000}$ | $8.00_{\pm 0.000}$ | $\mathbf{5.40}_{\pm 0.301}$ |
| Servo | $10.16_{\pm 1.650}$ | $14.88_{\pm 0.250}$ | $\mathbf{6.47}_{\pm 1.290}$ |
| Electrical | $12.00_{\pm 0.000}$ | $12.00_{\pm 0.000}$ | $\mathbf{8.06}_{\pm 0.201}$ |
| Supercond. | $34.14_{\pm 0.555}$ | $81.00_{\pm 0.000}$ | $\mathbf{12.57}_{\pm 0.081}$ |
| White Wine | $11.00_{\pm 0.000}$ | $11.00_{\pm 0.000}$ | $\mathbf{6.02}_{\pm 0.212}$ |
| Real Estate | $5.00_{\pm 0.000}$ | $5.00_{\pm 0.000}$ | $\mathbf{3.95}_{\pm 0.074}$ |
| Wind | $13.63_{\pm 0.048}$ | $13.00_{\pm 0.000}$ | $\mathbf{7.82}_{\pm 0.155}$ |
| CPU Activity | $12.00_{\pm 0.000}$ | $12.00_{\pm 0.000}$ | $\mathbf{9.50}_{\pm 0.193}$ |
| Echocardiogram | $7.36_{\pm 0.108}$ | $8.43_{\pm 0.099}$ | $\mathbf{5.23}_{\pm 0.101}$ |
| Iranian Churn | $9.15_{\pm 0.041}$ | $10.51_{\pm 0.061}$ | $\mathbf{4.87}_{\pm 0.161}$ |
| **Norm. Avg.** | $0.85_{\pm 0.009}$ | $0.94_{\pm 0.005}$ | $\mathbf{0.53}_{\pm 0.020}$ |

Table 10: Execution time comparison (in seconds – lower is better)

| Dataset | LIME | SHAP | MAPLE | BELLA |
|---|---|---|---|---|
| Auto MPG | $2.32_{\pm 0.040}$ | $0.06_{\pm 0.000}$ | $\mathbf{0.01}_{\pm 0.000}$ | $1.35_{\pm 0.030}$ |
| Bike | $1.42_{\pm 0.014}$ | $0.12_{\pm 0.001}$ | $\mathbf{0.01}_{\pm 0.000}$ | $3.24_{\pm 0.010}$ |
| Concrete | $2.26_{\pm 0.030}$ | $0.04_{\pm 0.000}$ | $\mathbf{0.02}_{\pm 0.000}$ | $0.92_{\pm 0.010}$ |
| Servo | $1.42_{\pm 0.061}$ | $0.18_{\pm 0.001}$ | $\mathbf{0.02}_{\pm 0.000}$ | $3.03_{\pm 0.112}$ |
| Electrical | $1.93_{\pm 0.001}$ | $0.11_{\pm 0.001}$ | $\mathbf{0.01}_{\pm 0.000}$ | $1.95_{\pm 0.004}$ |
| Supercond. | $21.65_{\pm 0.123}$ | $0.27_{\pm 0.004}$ | $\mathbf{0.02}_{\pm 0.001}$ | $243.51_{\pm 0.917}$ |
| White Wine | $3.92_{\pm 0.051}$ | $0.20_{\pm 0.001}$ | $\mathbf{0.02}_{\pm 0.000}$ | $1.70_{\pm 0.012}$ |
| Real Estate | $1.84_{\pm 0.084}$ | $0.02_{\pm 0.000}$ | $\mathbf{0.01}_{\pm 0.000}$ | $0.77_{\pm 0.030}$ |
| Wind | $2.24_{\pm 0.001}$ | $0.13_{\pm 0.001}$ | $\mathbf{0.01}_{\pm 0.000}$ | $1.45_{\pm 0.003}$ |
| CPU Activity | $1.87_{\pm 0.001}$ | $0.12_{\pm 0.001}$ | $\mathbf{0.01}_{\pm 0.000}$ | $1.66_{\pm 0.021}$ |
| Echocardiogram | $1.17_{\pm 0.001}$ | $0.02_{\pm 0.001}$ | $\mathbf{0.01}_{\pm 0.001}$ | $2.57_{\pm 0.532}$ |
| Iranian Churn | $3.00_{\pm 0.041}$ | $0.08_{\pm 0.001}$ | $\mathbf{0.02}_{\pm 0.000}$ | $1.40_{\pm 0.012}$ |

Table 11: Robustness comparison (0 to 1 – larger is better)

| Dataset | Number of Neighbours = 5 | | | | Number of Neighbours = 20 | | | |
|---|---|---|---|---|---|---|---|---|
| | LIME | SHAP | MAPLE | BELLA | LIME | SHAP | MAPLE | BELLA |
| Auto MPG | $0.90_{\pm 0.040}$ | $0.68_{\pm 0.083}$ | $0.70_{\pm 0.050}$ | $\mathbf{0.96}_{\pm 0.020}$ | $0.83_{\pm 0.024}$ | $0.58_{\pm 0.073}$ | $0.65_{\pm 0.047}$ | $\mathbf{0.93}_{\pm 0.028}$ |
| Bike | $0.69_{\pm 0.040}$ | $0.63_{\pm 0.030}$ | $0.47_{\pm 0.060}$ | $\mathbf{0.83}_{\pm 0.060}$ | $0.67_{\pm 0.047}$ | $0.62_{\pm 0.037}$ | $0.48_{\pm 0.050}$ | $\mathbf{0.85}_{\pm 0.048}$ |
| Concrete | $\mathbf{0.81}_{\pm 0.050}$ | $0.73_{\pm 0.050}$ | $0.76_{\pm 0.070}$ | $0.79_{\pm 0.040}$ | $0.67_{\pm 0.067}$ | $0.58_{\pm 0.046}$ | $0.65_{\pm 0.039}$ | $\mathbf{0.73}_{\pm 0.070}$ |
| Servo | $\mathbf{0.85}_{\pm 0.050}$ | $0.46_{\pm 0.040}$ | $0.63_{\pm 0.110}$ | $0.74_{\pm 0.040}$ | $\mathbf{0.80}_{\pm 0.033}$ | $0.59_{\pm 0.041}$ | $0.62_{\pm 0.1274}$ | $0.76_{\pm 0.1347}$ |
| Electrical | $0.82_{\pm 0.034}$ | $0.55_{\pm 0.037}$ | $0.62_{\pm 0.046}$ | $\mathbf{0.88}_{\pm 0.028}$ | $0.76_{\pm 0.036}$ | $0.50_{\pm 0.022}$ | $0.58_{\pm 0.028}$ | $\mathbf{0.88}_{\pm 0.026}$ |
| Supercond. | $0.88_{\pm 0.013}$ | $0.69_{\pm 0.044}$ | $0.68_{\pm 0.091}$ | $\mathbf{0.94}_{\pm 0.035}$ | $0.86_{\pm 0.072}$ | $0.73_{\pm 0.013}$ | $0.68_{\pm 0.018}$ | $\mathbf{0.95}_{\pm 0.067}$ |
| White Wine | $0.74_{\pm 0.070}$ | $0.58_{\pm 0.080}$ | $0.62_{\pm 0.070}$ | $\mathbf{0.87}_{\pm 0.060}$ | $0.66_{\pm 0.061}$ | $0.46_{\pm 0.033}$ | $0.53_{\pm 0.045}$ | $\mathbf{0.83}_{\pm 0.038}$ |
| Real Estate | $0.73_{\pm 0.080}$ | $0.71_{\pm 0.071}$ | $0.76_{\pm 0.059}$ | $\mathbf{0.94}_{\pm 0.060}$ | $0.66_{\pm 0.052}$ | $0.64_{\pm 0.068}$ | $0.67_{\pm 0.038}$ | $\mathbf{0.82}_{\pm 0.090}$ |
| Wind | $0.64_{\pm 0.043}$ | $0.61_{\pm 0.043}$ | $0.66_{\pm 0.022}$ | $\mathbf{0.99}_{\pm 0.009}$ | $0.68_{\pm 0.050}$ | $0.61_{\pm 0.041}$ | $0.61_{\pm 0.025}$ | $\mathbf{0.99}_{\pm 0.008}$ |
| CPU Activity | $0.58_{\pm 0.080}$ | $0.73_{\pm 0.060}$ | $0.67_{\pm 0.070}$ | $\mathbf{0.82}_{\pm 0.060}$ | $0.55_{\pm 0.095}$ | $0.66_{\pm 0.052}$ | $0.62_{\pm 0.051}$ | $\mathbf{0.79}_{\pm 0.044}$ |
| Echocardiogram | $0.83_{\pm 0.039}$ | $0.59_{\pm 0.040}$ | $0.52_{\pm 0.046}$ | $\mathbf{0.97}_{\pm 0.016}$ | $0.80_{\pm 0.036}$ | $0.57_{\pm 0.026}$ | $0.45_{\pm 0.031}$ | $\mathbf{0.96}_{\pm 0.032}$ |
| Iranian Churn | $0.77_{\pm 0.06}$ | $0.83_{\pm 0.020}$ | $0.70_{\pm 0.050}$ | $\mathbf{0.86}_{\pm 0.049}$ | $0.76_{\pm 0.058}$ | $0.77_{\pm 0.054}$ | $0.62_{\pm 0.066}$ | $\mathbf{0.84}_{\pm 0.048}$ |
| **Average** | $0.77_{\pm 0.050}$ | $0.65_{\pm 0.049}$ | $0.65_{\pm 0.062}$ | $\mathbf{0.88}_{\pm 0.040}$ | $0.73_{\pm 0.053}$ | $0.61_{\pm 0.042}$ | $0.60_{\pm 0.047}$ | $\mathbf{0.86}_{\pm 0.053}$ |

