# OpenReview forum: "BELLA: Black-box model Explanations by Local Linear Approximations"
_TMLR — Rejected by TMLR_

### Review · Reviewer_QUGn · 2024-11-08

**Summary Of Contributions:**

This paper proposes a method to generate factual and counterfactual explanations of a black-box model's output.

In a nutshell, given an example $x$ with some output value to be explained, the proposed approach first sorts the remaining training examples in terms of a given distance metric with respect to $x$. Then, with increasing neighbourhood size, it compute different local approximations of the black-box model around $x$, and evaluates them in terms of (i) size of the neighbourhood (for generality) and (ii) surrogate performance (for explanation accuracy). Then, it returns the best computed surrogate (for the chosen neighbourhood).

Overall, the paper is well organized and clearly written. The related work is appropriately studied, although the contribution (with respect to related works) remains quite unclear in the introduction. The considered desiderata for the explanations are appropriately defined and motivated in Section 3. The description of the proposed framework is overall sufficient, and the approach seems sound. It however sounds more like an engineering work, with the proposed approach essentially using existing concepts. It could still be an interesting contribution, but many notation and rigorousness issues must be fixed, and different aspects must also be clarified as detailed later in my review.

**Audience:**

Yes

**Broader Impact Concerns:**

Not sure if this part is necessary, but a short paragraph on the importance of the proposed method for XAI (in particular regarding the identified desiderata) and on its limits (in particular in terms of guarantees) may be worth it.

**Claims And Evidence:**

Yes

**Requested Changes:**

**Notation must be carefully checked to improve rigorousness and clarity. Among other issues, hereafter are some major inconsistencies:**

* according to the linear equation provided at the beginning of Section 4 (page 4), $n$ seems to be the number of features (it is also defined this way in Algorithm 2, and in page 6 after Algorithm 2). It is then used as the index for the distance measure between numerical features $d_n$ (on page 4), and, later, this same $n$ denotes the "sample size" on pages 5 and 6 (e.g., in (2)) and later in (5) (page 11).

* $i$ is used to index examples on page 6 (and in Algorithm 2) and 8 (through $x_i$) for instance. It is also used to index attributes in (1). Similarly, $j$ indexes attributes in (1) and examples in (2). While I understand that this is not technically wrong (since they are indices), it is good practice to keep using consistent indices for consistent entities.

* $x$  is used in most of the paper to denote an example (and defined this way at the beginning of Section 4), but then used to denote the value of an attribute (bottom of page 4) or a label (text after (2) on page 5). Similarly, $y$ is used to denote a label in most of the paper (e.g., in (2) or Algorithm 4) but is also used to denote an attribute value (bottom on page 4).

* $\delta^{ij}$ is defined as a function computing the distance between two values of a given attribute, while $\delta$  is later redefined in (2) to denote another measure.

* $\Delta$ is defined as a distance function between outcomes in (2), and then as a set of features that have to be modified between a given example and one of its counterfactuals in (4).

* $Y(x)$ is defined as the target value of a given sample at the beginning of Section 3, using a function notation. It is then defined as a set of labels (e.g., in Algorithm 1), which is not very rigorous.

* What is the set $S$ in (4) ? If it is the whole training set, why isn't it $T$ ? If it is the set of candidates with bounded distance to the reference value, why not match the notation of Algorithm 4 ?

**Others:**

* I found Figure 2 difficult to interpret. In particular, what do the two separate columns represent ?  If the right column is the continuation of the left one (as the dotted line seems to suggest), why is the y-axis the same on both ? Or is the right one the total positive contributions and the left one the total negative contributions ? It seems to no be the case since the coefficient associated to freqSMS is positive (and since it is a "number of SMS" as stated in the main text, how could the product be negative ?) Providing the attributes' values associated to the explained example would facilitate understanding.

* From a theoretical perspective, how does BELLA models/handles robustness ?

* For counterfactual explanations: the approach considers examples from the training set whose output by the black-box at hand is no further than $\epsilon$ from the target value. (i) what if there does not exist any such training example ? and (ii) can't this approach lead to counterfactual examples that are considerably further than counterfactual that may not belong to the training set (synthetic samples) ?

* Experiments on robustness: are the results similar when the number of considered neighbours is greater (or smaller) than 10 ? Could you provide additional explanations regarding the choice of these values ?

* It seems that the distance measure (1) does not take the label into account. How is that problematic (since explaining different predictions may require significantly differing features) or why isn't it ?

* In Section 5.1, could you elaborate on the choice of the computation for the target value ? (for counterfactuals generation)

* Section 5.1 states that "BELLA is the only one that can provide both factual and counterfactual explanations". However, couldn't other frameworks such as SLIM be used within Algorithm 4 to produce counterfactual explanations ? It is also unclear to me what guarantees are provided by Algorithm 4 ? Since the explanations computed by BELLA may not be 100% accurate, what if the proposed modifications do not lead to an actual outcome from the black-box within the desired target range ?

* Table 2 shows that the state of the art MAPLE framework is often better than the proposed BELLA in terms of fidelity. Since fidelity is only computed as the outcome difference between the black-box at hand and the trained surrogate for the explained example (or if I am wrong, please clarify this), couldn't this be improved by setting a larger weight to the explained example (than its neighbours) while training the surrogate ? Furthermore in this Table, I am not convinced that the computed fidelity for the counterfactual BELLA is meaningful (when evaluating counterfactuals, we usually set as a hard constraint the belonging to a desired target range, and rather evaluate the counterfactual's attribute values (e.g., in terms of distance from the explained example). In any case, comparing these values to the fidelity scores for factual explanations seems misleading and should be avoided (what do the bold values for the counterfactual BELLA stand for ?).

* In Table 3, generality is measured as the number of training examples that were considered to compute the explanation. Could you elaborate on that ? In particular, it is possible that some example belonging to the considered neighbourhood is not well explained, and in such case including it may not make sense ? Furthermore with comparison to LIME, why would evaluating fidelity on a finite number of points (discrete support) be better than on an hypercube (as done in LIME through range of values for each feature) ?

* In Table 5, the advantage of BELLA in terms of robustness is not consistent nor very clear. This relates to one of my previous points regarding how does BELLA handle robustness and how it could handle it better.

* Could you elaborate on the last paragraph on Section 5 (Verification on an interpretable model), possibly by providing some illustrative examples or more quantitative results in a dedicated Appendix section ?

* The fact that LIME's simplicity is set to the final simplicity of BELLA may result in unfair comparisons with respect to the other considered desiderata. Cross-validating the hyperparameters of LIME would result in a more thorough evaluation and comparison.

* The future directions mentioned in the conclusion are rather limited. Restricting the explanations to ensure they are realistic and provide insights regarding which actions an individual could take to modify its outcome from the model was already investigated in the literature and denoted as actionability. I would expect more insights regarding the limits of the proposed framework, its possible short-term improvements and longer-term perspectives.

* In the Appendix A (experiments with a Random Forest black-box rather than a Neural Network), why consider forests with $1,000$ trees (default being $100$ in scikit-learn, which is the used library) ?

* How does the proposed approach compare to the considered baselines in terms of computational time ? If there is any notable and consistent difference, it would be interesting to mention it and eventually add a few quantitative results.

**Minor:**

* Page 1: "eXplainable Artificial Intelligence"/"xAI" (why minuscule x ?)

* Page 4: "how their can be used" -> should be "they"

* Page 6: "the training easily lead to overfitting" -> "leads"

* Page 9, footnote: the URL for the third repository could be shortened as done for the two others

* Problem in reference "Hastie et al., 2009" (author Jerome H Friedman appears twice)

**Strengths And Weaknesses:**

**Strengths:**

* Overall, the paper is clear and appropriately structured

* The proposed approach is appropriate to tackle the problem at hand. It is sufficiently described, the designed choices make sense and the experiments overall show that the method effectively generates explanations that are competitive to the state of the art in terms of the considered desiderata. The experimental setup seems sound and the experiments are quite extensive

**Weaknesses:**

* The paper lacks rigorousness, especially for notations

* A few additional discussions would facilitate both understanding and assessment of the contributions (see the next textbox). Regarding this aspect, the amount of novelty seems rather limited

---

> ### Author Response · Authors · 2024-11-17
>
> Thank you for your detailed review! We are happy that you found our work well organized, the related work appropriately studied, the design choices well explained, the experimental comparison to competitors sound, extensive, and convincing, and our desiderata well
>  motivated. Our contribution is fulfilling these desiderata all together.
>
> We have renamed the variables so that **the same name is no longer used for different things in different sections**. In Eq 4, *S* is the number of  data points whose target value is in an ϵ-vicinity to the reference value.
>
> In **Figure 2**, blue upwards arrows are positive contributions, red downward arrows are negative contributions, and the final value is their sum. Thus, it has to be a minus in the equation on Page 7; we apologize for this oversight.
>
> **The distance measure does not take the label into account** because there can be data points with different feature values and a similar prediction (like sin(x)). Then BELLA provides the explanation for the “correct” local neighborhood.
>
> To **handle robustness**, we use Lasso regularization, which makes results less brittle. Also, our target metric increases number of data points, which makes the result more general and thus more robust.
>
> For robustness, since LIME and SHAP are less general than BELLA and MAPLE, and MAPLE's neighborhood is not crisp, more neighbors would actually benefit BELLA. Less neighbors will make everyone more robust. We will add these experiments.
>
> The **advantage of BELLA** in Table 5 is not consistent, but in the majority of cases, as well as on average, BELLA outperforms both competitors. BELLA does not beat each competitor on each desideratum, but it is **the only method that satisfies all boolean desiderata**, and, **in addition, it generally performs on par with, or better than, the competitors**.
>
> For counterfactual explanations, if there is **no training example in the neighborhood**, the neighborhood has to be enlarged, leading to more remote example points. That is the price to pay for the fact that BELLA only ever produces real examples, and **never synthetic examples**.
>
> To be realistic, the **target value** y’ cannot be too far away from y. Therefore, we vary y’ in the interval y+/- 30%.
> Any algorithm that produces a linear model and a neighborhood (SLIM) can be used in Algorithm 4 – but this excludes our direct competitors.
>
> If the proposed counterfactual modifications do not lead to an actual outcome from the black-box within the desired target range, one can reduce the impact of Lasso regularization to keep even the smaller feature modifications. Also, BELLA first selects the eligible candidates for the counterfactuals and then computes the smallest amount of change needed to reach the desired value. So, if the smallest change doesn't work, one can look at the proposed counterfactuals.
>
> We can indeed improve BELLA’s fidelity by giving more weight to the explained examples, but this would compromise the advantage of BELLA of delivering a linear explanation that is valid for the whole neighborhood with the same error margin.
>
> Factual and counterfactual fidelity appear in the same table mainly to show that they are both in the same order of magnitude, not to compare the values as such.
>
> If we exclude datapoints that are not well explained from the neighborhood, we would compromise the concept of generality (as in “This explanation applies to 20 other people who are similar to you, but not to 300 others who are also similar to you but where our explanation did not work”).
>
> We can give a **hypercube** for the size of the neighborhood, but it is the number of data points (and not the size of a potentially sparse hypercube) that conveys the significance of the explanation.
>
> As you propose, **we will provide examples from our dataset** to illustrate the last paragraph on Section 5.
>
> One can indeed always tune simplicity until LIME beats BELLA on fidelity, and tune fidelity until LIME beats BELLA on simplicity. To compare the two methods, however, one has to fix one parameter and compare the other. This is what our experiments do: at the same simplicity, BELLA beats LIME on fidelity. It follows that, to achieve the same fidelity as BELLA, LIME necessarily has to decrease its simplicity. Thus, BELLA beats LIME in both cases.
>
> We will add an experiment with random forests with 100 trees, but we do not expect the results to change much, because BELLA works on the labeled dataset, no matter how it was produced.
>
> For runtime, BELLA is a factor of 3 slower than LIME, which is itself a factor of 10 slower than SHAP. We will add a table with the numbers.
>
> **Future work** can be the speed optimization of BELLA (e.g., by using kd-trees in Algorithm 1). One could also give up determinism and resort to sampling to further speed up the algorithm.
>
> We shall add all of these reasonings to the paper. We believe that clarifying the points you raised will make the advantages of BELLA even clearer.

---

> > ### Comment · Reviewer_QUGn · 2025-02-03
> >
> > Thank you for your detailed response and for the performed modifications, addressing most of my concerns. I especially appreciate the added experiments varying the size of the considered neighbourhood, reporting computation times, and the new Figure 3 highlighting BELLA's consistence with an interpretable model. I hereafter review my remaining concerns (note that most of them are simple advices or comments) along with some recommendations.
> >
> > - the fact that $n$ is used for different purposes at different locations is still a bit disturbing to me, but as it is well locally introduced I won't argue for that
> >
> > - regarding my question on robustness, I was talking about theoretical robustness guarantees, such as worst-case guarantees in a given neighbourhood defined in terms of some distributional or sample-based distance. I guess the approach of BELLA is more empirical, and since it still makes sense, I won't argue neither although I think it would be interesting to elaborate on that. At the end, I guess the empirical guarantee that BELLA could provide is something like "This explanation applies with an error at most XX in a YY-examples neighbourhood within the training set" ?
> >
> > - the fact that BELLA is limited to real examples can be seen as a strength, but it can also be a significant weakness, as one can expect a good explanation framework to be able to characterize the rationale behind a black-box model's decisions wherever they lie in the feature space. This may be especially true when the context of deployment differs from the training distribution, which often happens in practice. Couldn't BELLA be "hybridized" to handle these scenarios (e.g., by deciding to artificially populate an example's neighbourhood if its neighbourhood in the training dataset was not properly represented) ? Also, the fact that BELLA provides real examples might have privacy implications ? Maybe a few more words on that would be interesting in Section 6.
> >
> > - I still think that providing fidelity results for both factual and counterfactuals in the same table can be a bit misleading since these values can not be compared. I won't argue for that but a few words to warn the reader when referring to these results might be worse it
> >
> > I also share some of the other reviewers' concerns on the applicability of BELLA, regarding both:
> > - the underlying assumption that the training set is known, although this assumption is shared by other approaches in the literature and is somehow acceptable if we want to provide real examples as explanations
> > - the approach's scalability, since, as illustrated with the results on the "superconductivity" dataset, when the number of features increases, BELLA's runtime also increases significantly, in particular with respect to existing methods

---

> > > ### Author Response · Authors · 2025-02-04
> > > **Thank you!**
> > >
> > > Thank you for your detailed reply! We are happy that most of your concerns have been addressed!
> > >
> > > In what concerns the remaining points:
> > > - BELLA can indeed provide an “empirical guarantee” of the form you suggest. We can compute the error on the neighborhood, and then report that “This explanation applies with an error at most XX in a YY-examples neighborhood within the training set”, as you say.
> > > - As you summarize, BELLA works on the data itself, not on the data probed from the model, and this is useful in some scenarios, and maybe less useful in others. However, one can indeed probe the model to provide more data. This would give the “hybrid BELLA” you are suggesting, and that is in principle possible. We can state this in the paper. (Note that the inverse is much harder: if some approach needs the model, and one only has the data, then a hybrid solution is more difficult.)
> > > - As for the privacy implications in counterfactual examples: We do not actually return the target datapoint when giving a counterfactual explanation, but the original datapoint that has been modified. So we don't give away other datapoints. Nonetheless the modifications we propose might reveal information about that other datapoint. That is indeed a privacy implication we shall mention in Section 6!
> > > - We will add a sentence warning the reader that the numbers in the last column of Table 2 are not comparable.
> > >
> > > In what concerns the runtime optimization of BELLA, this is indeed an issue that we did not concern ourselves with. Our main contribution is (1) the definition of new desiderata for explanations and (2) a method to fulfil these desiderata. And even though BELLA runs just fine on the majority of datasets, the optimization of the algorithm remains an important item for future work, in particular for datasets with many attributes. We mention this in the conclusion and also in the newly added “Limitations” section, and we can highlight that the issue is particularly important for datasets with many attributes.

---

> > > > ### Comment · Reviewer_QUGn · 2025-02-04
> > > >
> > > > Thank you for your reply, answering all my comments!

---

### Review · Reviewer_pqpw · 2024-11-10

**Summary Of Contributions:**

The paper proposes Black-box model Explanations by Local Linear Approximations BELLA, a post-hoc explanation method for tabular data regression models. The paper aims to create an explanation method that satisfies the following criteria : *Accurate (fidelity):* reflect the predictions of the black-box model. *Simple:* provide an explanations using few features. *Robust* similar data points have similar explanations. *General* the number of data points to which an explanation applies is large. *Counterfactual* provides a set of modifications to the data point at hand that would entail a change in the decision of the black-box model. *Deterministic* the explanation for a given point is always the same if you run multiple times (no randomness). *Verifiable* the predicted value can be compute from the feature values.

## Method:

- **Given:** a tabular dataset $T$ with features set $F_i$ where $i$ is the number of features. A model $Y$ (this is the black box model to be explained) produces an output $y$ (target value) given an input $x \in T$, i.e. $Y(x)=y$.

- **Goal** find a linear equation $y ≈ w_1·f_1+w_2 ·f_2+· · ·+w_n·f_n+w_0$, where wi are real-valued regression coefficients and $f_i$ are feature values of $x$ in $T$.

_____________

**To achieve this Goal, Bella has 3 main steps:**

### 1- Compute the distance of $x$ to the other points in $T$

All numerical features are normalized to be on the same scale; the generalized distance function is used to account for numerical, categorical, and binary data types. Where the distance for numerical features is the L1 norm, for categorical features, the distance proposed by (Ahmad & Dey, 2007) is used, which takes into account the distribution of values and their cooccurrence with values of other attributes; for binary features, the Hamming distance is used. The overall distance is given in Equation 1.

### 2- Neighborhood Search to find samples in the training dataset $T$ that is closest to the input $x$.

To get an appropriate neighborhood, both the distance between the training sample and the target, as well as the size of the neighborhood, are considered in the optimization function. They employ a linear search algorithm, as given in algorithm 2. They sort the training set by increasing distance to $x$, train a linear model on the first $i$ data points for increasing $i$, and return the set of neighbors for which the lower bound of the Berry-Mielke universal R value is maximal. They proposed to use Berry-Mielke as it measures how much better the model is compared to a random model and it is generally easy to interpret.

### 3- Building a local surrogate model

A linear model is trained on the neighborhood selected in step 2. Highly collinear features are removed by computing the variance inflation factor (VIF) and setting a threshold as the cut-off value for the VIF,  Lasso regularization is used for feature selection (here Lasso is not used for prediction), and finally, an Ordinary Least Squares linear model on the subset of variables selected by Lasso.

_____________

- **Output** BELLA outputs the OLS model computed in step 3 together with the size of the neighborhood selected in step 2.

- **BELLA explanations** Given the above steps BELLA explanations satisfy the above criteria except for counterfactuality. For counterfactuality, since this is a regression scenario, the goal is to suggest changes to the current input to reach the value that the user would like to see. To provide such an explanation, the paper proposes the following:

  - Select data points whose target value is in an ϵ-vicinity to the reference value

  - Optimize two criteria: the distance between the given data point and the candidate, and the amount of change needed.

  - They propose minimizing the average amount of change and suggesting smaller adjustments to multiple features rather than large changes for small number of features optimizing objective in equation 3 using algorithm 4.

## Experiments:

- **Datasets** Experiments were preformed on 12 tabular datasets with feature space ranging from 4-81 and training dataset ranging from ~ 167 up to 21K.

- **Models** explanation methods was applied to a random forest and a single-layer MLP.

- **Baselines** Bella is compared to LIME, SHAP and MAPLE.

- **Results**

  - *Fidelity* MAPLE is the best, but BELLA is comparable.

  - *Counterfactuality* the fidelity of the counterfactual explanations produced by BELLA with respect to a reference value is comparable to the factual explanations produced by LIME.

  - *Generality* they use the number of data points to which the explanation applies to in the training dataset as a measure of generality overall BELLA has higher number of samples per explanation.

  - *Simplicity* the number of features that an explanation contains on average, BELLA produces simpler explanations compared to SHAP and MAPLE.

  - *Robustness* this is how similar the explanations for close data points are, to measure robustness they compute explanations for each data point in the test set, and compute robustness with respect to the 10 closest data points using equation 5. On average BELLA provides more robust explanations.

**Audience:**

Yes

**Broader Impact Concerns:**

This paper is more of a statistical or classical machine learning paper (both in terms of the method and the way the paper is written). Overall, I do not think BELLA can be practically used as a post-hoc explanation method for modern deep-learning models that have thousands of features and are trained on hundreds of thousands of examples.

**Claims And Evidence:**

Yes

**Requested Changes:**

- Typo on page 5 in the paragraph starting with Step 2: Neighborhood Search; typo is *metrics metrics*.

- In the experiment section the size of the test dataset for which different results were shown was not mention (unless it is instances in table 1) please clarify.

- The paper is saying the same thing about counterfactuality in the last paragraph in page 9 as it says in the first paragraph in page 12 please do not repeat the same information twice.

- This is not a request but more of a suggestion; you can formulate Bella as an interpretable framework for tabular model prediction and compare its performance to standard black box models, and if it is comparable, you can suggest replacing Bella with the black box in many cases a mixture of simple linear models for tabular data can be quite powerful [1]. Basically, suggest Bella as a replacement for black box models for tabular data in small data regimes.

[1] Ismail, Aya Abdelsalam, et al. "Interpretable mixture of experts." arXiv preprint arXiv:2206.02107 (2022).

**Strengths And Weaknesses:**

# Strengths:

- The paper is well motivated and well written. For example the quality measures proposed for a good explanation method in section 3 are extremely clear and can be used as a standard of what a good explanation method should accomplish.

- The reasoning for each step in the method is clear and very detailed (which is actually quite uncommon in ML papers, so it was great to see this).

- Bella satisfies all criteria one would hope to see in an explanation method, the paper also suggests a way to use bella to generate counterfactuals which can be very useful for real world applications.

- Bella can actually be used instead of the black-box model rather than a post-hoc explanation method, if given the output as the ground truth target instead of the black box predictions.

# Weakness:

 - Bella is only suitable for tabular data.

- Bella assumes that the training dataset which the model was trained on is known when we want to explain that model which is often not the case.

- The method is very computationally expensive the cost of explanation can be higher than training the prediction model itself.

- When training dataset becomes large for example a few hundred thousands samples or feature space becomes large for example a few thousands (which is very typical in current ML setups) applying Bella becomes infeasible .

- The results show that BELLA is more general, robust, and simpler than other baselines but the difference is not significant compared to the computation cost the other have especially if one would use gradient SHAP as an approximation of SHAP for example. So its very difficult to justify the use of BELLA over other baselines given the computational cost.

---

> ### Author Response · Authors · 2024-11-17
> **Thank you for your review!**
>
> We would like to thank you for your detailed review (and in particular for the very accurate summary of our contributions). We are delighted that you found the paper well motivated and well written, that you highlighted in particular the clarity of our reasoning steps, and that you appreciate our main contribution: that **BELLA satisfies all desiderata that one would hope to see in an explanation method**.
>
> We concur with you that the **input of BELLA is a tabular dataset**. Note, however, that the competing methods LIME and SHAP need not just this dataset, but in addition also the model itself (so that they can probe it). Thus, BELLA actually has less requirements than LIME and SHAP. A positive side effect of this is that, different from SHAP and LIME, BELLA can explain *any* labeled dataset, be it by a model or from another source (e.g., housing prices from real estate data).
>
> You are worried about the **computational cost of BELLA**. For each point that is to be explained, BELLA has to find the 2n closest points (Algorithm 1), where n is the number of features. Then BELLA trains 2n local linear models (Algorithm 2) with maximally n parameters on maximally 2n points.
> * You wonder how this complexity compares to the models we seek to explain. This depends on the number of features, the number of data points, and the complexity of that model, and thus has no general answer. However, the models we seek to explain typically have not O(n) parameters, but typically hundreds if not thousands of parameters (see Section 5/Datasets), and are trained not on 2n data points, but on all data points (of which there are usually orders of magnitudes more than n, see Table 1). We shall mention these considerations in the paper.
> * You wonder how this complexity compares to BELLA’s competitors. BELLA is indeed more computationally expensive than the other models. This is because, different from our competitors, BELLA is deterministic. Our method thus has to explore the whole local space.
> * Finally, you wonder about the general applicability of BELLA. We agree that the number of features and number of data points will affect the runtime of BELLA. That said, the datasets we use are the standard benchmarks in the domain, on which also our competitors are evaluated. They are of decent size, with dozens of features and thousands of instances. Here, BELLA runs just fine (in a matter of seconds per datapoint), **in the same order of magnitude as LIME**. We shall report detailed numbers in the paper.
>
> Generally, **we did not focus on efficiency but on defining and achieving desiderata of explanations**. We show that BELLA is the only method that satisfies all boolean desiderata (deterministic, verifiable, and both factual and counterfactual explanations), and that, in addition, it generally performs on par with, or even better than the competitors across the other quality measures. For future work, the implementation of BELLA can indeed be sped up (e.g., by using kd-trees for computing the neighborhoods). We shall point out this trade-off in the paper.
>
> The **typo** and repetition you mentioned are well received and will be removed. The size of the testing dataset is 10%, and we will mention this.

---

### Review · Reviewer_R3TT · 2025-01-06

**Summary Of Contributions:**

The paper introduces BELLA as a deterministic, model-agnostic approach for explaining regression-based black-box models (or even arbitrary numeric variables in tabular datasets) via local linear approximations. The authors motivate the need for explanations that are not only accurate, simple, robust, and general, but also deterministic and capable of delivering counterfactuals. BELLA addresses these requirements by (1) defining a crisp neighborhood around a point of interest in the original feature space, (2) training a sparse linear surrogate in that neighborhood to produce factual explanations, and (3) producing counterfactual suggestions by searching among data points whose target lies close to a desired reference value. The paper’s experimental results across a variety of datasets and black-box models (random forests and neural networks) suggest that BELLA offers a compelling mix of simplicity, generality, and robustness, while maintaining competitive fidelity to the original model.

**Audience:**

Yes

**Broader Impact Concerns:**

While BELLA aims to enhance interpretability and trust in complex models by providing deterministic local explanations, several broader ethical considerations remain. First, explanatory bias could arise if the underlying black-box model itself encodes unfair or discriminatory patterns; BELLA’s explanations, being local approximations, do not necessarily correct for or identify such biases. Second, there is a risk of over-trust: users might conflate local linear surrogates with ground-truth rationale, ignoring that these explanations only approximate the model's decision boundary around a single point (or neighborhood). In high-stakes domains (e.g., healthcare, legal, or credit scoring), such misinterpretations could lead to complacency or insufficient oversight. Third, because BELLA relies on existing training data for counterfactual suggestions, data quality issues such as historical discrimination or sampling biases may propagate into suggested actions, thus reinforcing existing inequities if not critically assessed. Finally, manipulation is conceivable if individuals learn how to exploit local explanations to “game” the model without genuinely improving underlying attributes (e.g., artificially adjusting specific features to increase a credit score). These concerns underscore the importance of combining BELLA with frameworks for fairness, transparency, and accountability, as well as thorough domain-specific audits.

**Claims And Evidence:**

Yes

**Requested Changes:**

(Provided in the previous section)

**Strengths And Weaknesses:**

Strengths : A key strength lies in the clear objective of maximizing both accuracy and size of the local neighborhood, formalized by a non-trivial use of the Berry-Mielke universal R value and confidence intervals to guide the neighborhood search. This balances the trade-off between fidelity and generality, rather than heuristically focusing on one aspect. The method’s determinism, achieved by avoiding synthetic data generation or random sampling, ensures that the same input consistently yields the same explanation, making BELLA more transparent for end-users who need repeatability. Another important contribution is the introduction of both factual and counterfactual modes; BELLA’s ability to produce a sparse linear model is extended seamlessly with a systematic search among “candidate” points whose target is near the desired outcome, thus providing actionable insights into how the original features can be changed. The authors also highlight “verifiability” by explicitly returning a linear function, so an analyst can plug in any set of feature values in the neighborhood to check how the method predicts the outcome—this is rarely seen in approaches such as LIME or SHAP, which rely on feature projections that do not directly map to the original input space.


Weaknesses: While BELLA offers significant advantages, certain technical aspects merit refinement. First, the counterfactual explanation process relies heavily on existing data points whose target values lie near the desired reference, implicitly assuming that “realistic” neighbors exist in the training set. However, if a user-specified target value is sufficiently far from all data points, BELLA may produce no viable counterfactuals, or produce ones based on unmodifiable features (e.g., age, medical history). A more domain-driven mechanism that enforces feasibility constraints or flags unchangeable features (e.g., by re-weighting the distance metric or integrating specialized optimization layers) would enhance the method’s practical relevance. Second, the linear search for the “best neighborhood size” can be computationally expensive on large datasets since it requires retraining a linear model (and re-computing Berry-Mielke’s
𝑅 R confidence bounds) for each candidate size. Some form of incremental fitting strategy or binary search could reduce overhead while still locating an appropriate neighborhood. Third, although the unified distance function for numerical, categorical, and binary features is flexible, the paper does not investigate how domain-specific weighting or dynamic scaling of feature distances might affect performance, particularly in cases with highly imbalanced categories or strong collinearity. Finally, BELLA’s linear surrogate may fail to capture intricate, higher-order interactions within the black-box model, especially in high-dimensional or extremely non-linear settings, so further research on locality-aware non-linear surrogates or piecewise-linear decompositions could strengthen the model’s ability to explain more complex behaviors.

---

> ### Author Response · Authors · 2025-01-10
> **Thank you for your review!**
>
> Thank you for your detailed review! We are happy that you find that “BELLA offers a compelling mix of simplicity, generality, and robustness, while maintaining competitive fidelity to the original model”! This is our contribution exactly!
>
> Thank you also for appreciating BELLA’s advantages in detail: its determinism, our formalization, BELLA’s transparency, and in particular our criterion of verifiability, which is indeed, as you say, rarely seen in competing approaches.
>
> You are raising the problem of **producing impossible counterfactuals**. Finding good counterfactuals is indeed an open problem. That said, BELLA is the only approach among current state-of-the-art approaches that can deliver both factual and counterfactual explanations at all. Fine-tuning the counterfactuals is, as we say in our conclusion, left for future work. This can include, as you suggest, domain-specific constraints. We shall make this explicit.
>
> You are also raising the issue of **computational complexity**. BELLA comes indeed with a higher computational cost, which is the price we have to pay for determinism. That said, the datasets we use are the standard benchmarks in the domain, on which also our competitors are evaluated. They are of decent size, with dozens of features and thousands of instances. Here, BELLA runs just fine (in a matter of seconds per datapoint), in the same order of magnitude as LIME. We shall report detailed numbers in an updated version of the paper.
>
> You regret that we do not investigate how **domain-specific weighting or dynamic scaling of feature distances** affects performance. This is for sure an interesting question. However, in all fairness, competing approaches do not investigate this either, and it would go beyond the scope of this paper to do so. That said, we shall mention it as a question that merits investigation in future work.
>
> You worry that BELLA’s linear surrogate may fail to capture intricate, **higher-order interactions within the black-box model**. This problem is part of the intrinsic trade-off between simplicity and accuracy, which any post-hoc explainability approach finds itself in. We shall mention it explicitly in the paper.
>
> You rightly point out that **BELLA does not correct for biases in the data or model**. That is true, but it is a feature rather than a bug: BELLA’s explanations make such biases visible to the user. It would be disastrous to correct them and thus convey the impression that the model does not have them. The same goes for data quality issues such as historical discrimination or sampling biases. We shall point this out in the paper.
>
> You mention the problem of **overtrust**, where users mistakenly take a local explanation as a global one. That is an interesting issue (albeit inherent in all local approaches), which we shall mention in the paper.
>
> You mention the possibility that users **“game” the model** without genuinely improving underlying attributes (e.g., artificially adjusting specific features to increase a credit score). That, however, appears to be a problem with the model, not with the explanation.
>
> Overall, we agree with your assessment that BELLA, as any explainability method, gains from being combined with frameworks for fairness, transparency, and accountability, as well as thorough domain-specific audits. We shall mention this in the paper.
>
> *We will submit a revised version of our paper soon.*

---

### Author Response · Authors · 2025-03-17
**Current status**

Hi,

we would like to thank the reviewers again for their thoughtful reviews. We believe we have addressed their concerns in our new version of the paper (submitted on January 22nd). If there are any further steps needed from our side, please let us know!

Thank you,

the authors

---

### Decision · Action_Editor_9ACH · 2025-03-20

**Recommendation:** Reject

**Comment:**

I recommend rejection at this time due to the weak support for the counterfactual explanations claim, as discussed in the Claims and Evidence section above. (I recognize that this recommendation may be more disappointing than usual because of the abnormally long time it took to review this submission by TMLR standards.)

The comments about counterfactual explanations under Claims and Evidence could be addressed by more experimental results and comparisons, or by weakening or even removing the claims. In addition, I also suggest the following smaller revisions:
- Generality: I second Reviewer QUGn's comment that the number of points used to fit the linear model might not be the best measure of generality if the fit is poor for some of these points, even though the fit is good in an average sense. Perhaps a notion of "effective number of points" could be considered.
- Computational complexity: The submission could be more explicit in acknowledging that the difficulty is with handling high feature dimensions $d$ (assuming that my understanding is correct).

**Audience:**

Two of the reviewers felt that the submission is of interest to the explainable AI community. Although the novelty of the linear surrogate model approach is limited, the work is well-motivated in seeking to achieve multiple desiderata simultaneously, and it thus complements the literature in the area.

Reviewer pqpw remained concerned after the rebuttal about the practicality of BELLA due to its higher computational cost. As I understand it, for $d$-dimensional features, BELLA potentially requires $O(d)$ linear regressions, each with up to $d$ parameters on $O(d)$ data points. Thus, I agree that BELLA currently does not scale well with feature dimensionality (the Superconductors row in Table 6, where $d=81$, already hints at this, although the addition of Table 6 to the manuscript is a positive overall). Nevertheless, I think that BELLA's approach (of satisfying multiple desiderata) is interesting enough even in the setting of low-to-moderate $d$.

**Claims And Evidence:**

This submission proposes a post hoc explanation method called BELLA for black-box regression models. BELLA aims to achieve multiple desiderata simultaneously, including generality (applicability to many input points), robustness (also with respect to input points), determinism, and counterfactuality (provision of counterfactual explanations), in addition to the usual fidelity and simplicity.

On the positive side, most of the above desiderata are demonstrated empirically. Reviewer QUGn had some concerns and questions regarding generality and robustness, which were mostly addressed by the rebuttal and by varying the number of neighbors used to evaluate robustness (but please see the Comments section below for a follow-up comment).

Regarding clarity, Reviewer QUGn noted many instances where mathematical symbols were overloaded or used inconsistently. These were nearly all addressed in the revised manuscript.

The weakest claim in my view is the one regarding counterfactual explanations. I think what is needed for this claim is convincing evidence that the counterfactual explanations produced are "of interest," i.e., that someone might consider using, particularly in light of existing counterfactual explanation methods. I think the submission falls short on this front.
- Reviewers R3TT and QUGn were both concerned that constraining counterfactual explanations to be close to training examples may overly sacrifice quality, in terms of criteria such as closeness to the given factual example. The authors responded that this may be the price to pay for BELLA's counterfactual explanations being "real" training examples. However, it is not quite true that the counterfactual explanations are "real" because the returned counterfactual $\hat{x}$ is a modification of a training example $x'$ (see e.g. Algorithm 4, line 4).
- Reviewer QUGn also found it odd to evaluate BELLA's counterfactual explanations in terms of fidelity and to show these fidelity values in the same Table 2 as for factual explanations (in spite of the note added to the manuscript that these values are not directly comparable).
- Building on the above comment, I find the evaluation of BELLA's counterfactual explanations incomplete, in that they are evaluated only in terms of fidelity, and the distances to the given factual examples $x$ are not reported.
- Perhaps more importantly, there is no comparison to state-of-the-art counterfactual explanation methods to see whether BELLA's counterfactuals would indeed be "of interest" (i.e., not too much worse).
- As noted by reviewers, BELLA does not consider other criteria such as actionability that existing counterfactual explanation methods do.

**Resubmission Of Major Revision:**

The authors may consider submitting a major revision at a later time.